# TRIM28 protects TRIM24 from SPOP-mediated degradation and promotes prostate cancer progression

Ka-wing Fong[1], Jonathan C. Zhao[1], Bing Song[1], Bin Zheng[1] & Jindan Yu[1,2,3]

TRIM24 is an effector substrate of the E3 ubiquitin ligase adaptor SPOP and becomes stabilized in prostate cancer (PCa) with SPOP mutations. However, how TRIM24 protein is regulated in the vast majority of SPOP-wildtype PCa is unknown. Here we report TRIM28 as a critical upstream regulator of TRIM24. TRIM28 protein interacts with TRIM24 to prevent its ubiquitination and degradation by SPOP. Further, TRIM28 facilitates TRIM24 occupancy on the chromatin and, like TRIM24, augments AR signaling. TRIM28 promotes PCa cell proliferation in vitro and xenograft tumor growth in vivo. Importantly, TRIM28 is upregulated in aggressive PCa and associated with elevated levels of TRIM24 and worse clinical outcome. TRIM24 and AR coactivated gene signature of SPOP-mutant PCa is similarly activated in human PCa with high TRIM28 expression. Taken together, this study provides a novel mechanism to broad TRIM24 protein stabilization and establishes TRIM28 as a promising therapeutic target.

[1] Division of Hematology/Oncology, Department of Medicine, Northwestern University Feinberg School of Medicine, Chicago, IL, USA. [2] Department of Biochemistry and Molecular Genetics, Northwestern University Feinberg School of Medicine, Chicago, IL, USA. [3] Robert H. Lurie Comprehensive Cancer Center, Northwestern University, Chicago, IL, USA. Correspondence and requests for materials should be addressed to J.Y. (email: jindan-yu@northwestern.edu)

Cancer genome characterization has recently revealed recurrent missense mutations in the Speckle-type POZ protein (SPOP) gene in 11–13% of primary prostate cancer (PCa)[1,2] and to a less 6–8% in metastatic, castration-resistant prostate cancers (CRPC)[3,4]. SPOP is the substrate-binding member of the E3 ubiquitin-protein ligase complex that mediates ubiquitination and proteasomal degradation of target proteins. It contains a BTB domain, which serves as an adapter for Cullin-based E3 ubiquitin ligase, and a MATH domain that is responsible for substrate recognition and CUL3-mediated protein degradation[5–7]. All PCa-associated SPOP mutations discovered thus far affect evolutionarily conserved residues within the MATH domain and alter its ability to bind substrates[8–10]. Through forming heterodimers with wild-type SPOP, SPOP mutants reduce wild-type-SPOP binding to substrates, resulting in dominant-negative effects on substrate binding, ubiquitination, and degradation[10]. To date, a large number of SPOP substrates have been discovered, including TRIM24 (tripartite motif 24 protein), DEK, ERG, SRC3, androgen receptor (AR), SENP7, and BRD4[8–16].

TRIM24, also known as TIF1α, contains an N-terminal tripartite motif (TRIM), comprised of a RING (E3 ubiquitin ligase domain), a B-box type 1 and 2 (B1B2), and a coiled-coil region (BBC), and a C-terminal PHD-Bromo dual epigenetic reader domain. Distinct from other TRIM proteins, TRIM24 harbors an evolutionarily conserved LxxLL motif in the middle domain, next to the PHD-Bromo domain, that interacts with the AF-2 domain of several ligand-dependent nuclear transcription factors, including AR[17,18]. As a substrate of SPOP-mediated degradation, TRIM24 protein is stabilized in the context of SPOP mutations, leading to enhanced AR signaling and cell growth[19]. Interestingly, TRIM24 protein and activities are elevated much more broadly than SPOP mutations in CRPC, suggesting additional mechanisms to TRIM24 upregulation and/or stabilization that may be particularly important to CRPC.

Tripartite motif-containing 28 (TRIM28), also known as TIF1β and KAP1, contains an N-terminal TRIM and C-terminal PHD-bromo domains similar as TRIM24. As a RING domain protein, TRIM28 has been shown to target p53 and AMPK for ubiquitination and degradation through proteasome-dependent pathways, promoting tumorigenesis[20,21]. TRIM28 is also a critical regulator of DNA damage response and colocalizes with many DNA damage response factors at sites of DNA strand breaks[22]. Moreover, TRIM28 has been shown to interact with ligand-dependent corepressor (LCoR), SETDB1, and HDAC1 to facilitate transcriptional repression[23,24]. In agreement with this, TRIM28 was found to be depleted from open chromatin and enriched in tumor-specific closed chromatin in prostate cancer cells[25]. TRIM28 has also been shown to interact with AR and induce AR activity in a reporter assay[26]. Human Protein Atlas Database showed that TRIM28 expression is relatively high in some cancers, including PCa, but low in others[27]. TRIM28 expression and function in PCa, however, have not been carefully examined.

Here we demonstrate that TRIM28 protein interacts with TRIM24 to prevent it from SPOP-mediated ubiquitination, thereby enhancing TRIM24 protein stability and expression levels. Further, we explored how TRIM28 facilitates TRIM24 and AR signaling and the significance of this regulatory pathway in clinical samples and during PCa tumorigenesis.

## Results

### TRIM28 is a positive regulator of TRIM24 protein stability.
TRIM24 is a substrate of SPOP and is stabilized in SPOP-mutant PCa[10,19]. And yet, TRIM24 protein is broadly upregulated in CRPC, even in those with wild-type SPOP, suggesting other essential regulatory pathways[19]. Indeed, western blot analysis showed strong TRIM24 expression in a panel of PCa cell lines that are SPOP wild type (Fig. 1a). To explore potential cofactors that might stabilize TRIM24, we performed tandem affinity purification combined with mass spectrum analysis of TRIM24-containing complexes in LNCaP cells and identified a number of proteins, ranked among top of which were TRIM family proteins TRIM28 and TRIM33 (Supplementary Table 1). To determine which of these interacting proteins might regulate TRIM24 expression, we performed shRNA screening of the top 10 TRIM24 interactors (Supplementary Fig. 1a–b). Immunoblotting demonstrated that knockdown of TRIM28, but not other inter-actors, remarkably reduced TRIM24 protein level (Fig. 1b). In agreement with this, western blot analysis showed positively correlated protein levels of TRIM24 and TRIM28 in a panel of prostate cell lines, including androgen-dependent PCa (ADPC) as well as CRPC (Fig. 1a). Concordantly, TRIM28 knockdown reduced TRIM24 protein levels in both androgen-depleted and androgen-stimulated cells, suggesting an AR-independent mechanism generally applicable to ADPC and CRPC (Fig. 1c). Similar results were confirmed using another independent shRNA targeting TRIM28 (Supplementary Fig. 1c) and was further elucidated in several additional PCa cell lines (Fig. 1d). By contrast, TRIM24 knockdown did not alter TRIM28 protein levels (Supplementary Fig. 1d).

Next, to understand the molecular mechanisms by which TRIM28 increases TRIM24 protein levels, we first performed qRT-PCR analysis and found that TRIM28 knockdown did not decrease TRIM24 mRNA, precluding transcriptional regulation (Fig. 1e). This, along with the fact that TRIM28 and TRIM24 proteins interact, suggests a potential mechanism involving post-transcriptional regulation. To examine whether TRIM28 increases TRIM24 protein stability, we treated control and TRIM28-knockdown LNCaP cells with cycloheximide, an inhibitor of new protein biosynthesis, to monitor the pace of protein degradation. Western blot analysis showed that TRIM24 was relatively stable in LNCaP cells with half-life greater than 12 h, which was however shortened to approximately 4 h in TRIM28-depleted cells (Fig. 1f, g), suggesting TRIM28 as a positive regulator of TRIM24 protein stability.

### TRIM28 stabilizes TRIM24 through protein–protein interaction.
To validate our mass spectrometry data, we performed immunoprecipitation of endogenous TRIM28 in LNCaP cells and indeed observed TRIM24 protein in the complex (Fig. 2a). Further, we overexpressed SFB (S-protein, Flag, and a streptavidin-binding peptide)-tagged TRIM24 in 293T cells and performed co-immunoprecipitation (co-IP) of ectopic SFB-TRIM24 in an S-beads pull down assay. Western blot analysis confirmed ectopic TRIM24 expression and pull-down, while also detecting endogenous TRIM28 as an interacting partner (Fig. 2b). In a reciprocal experiment, S-beads pull-down of ectopic TRIM28-SFB in 293T cells similarly retained TRIM24, supporting physical interaction between TRIM28 and TRIM24 proteins (Fig. 2c).

As TRIM family proteins contain multiple functional regions including RING, B1B2, BBC, HP1, and PHD/Bromo domains[28–30], we next sought to determine which of these domains mediate the interaction between TRIM28 and TRIM24. We first generated a series of SFB-tagged TRIM24 truncation mutants and transfected them into LNCaP cells, but had difficulty in expressing several constructs, potentially due to proteasome-mediated degradation since MG132 was able to restore some expression (Supplementary Fig. 2a, b). To circumvent this issue, we generated a series of GST-tagged TRIM24 truncation mutants.

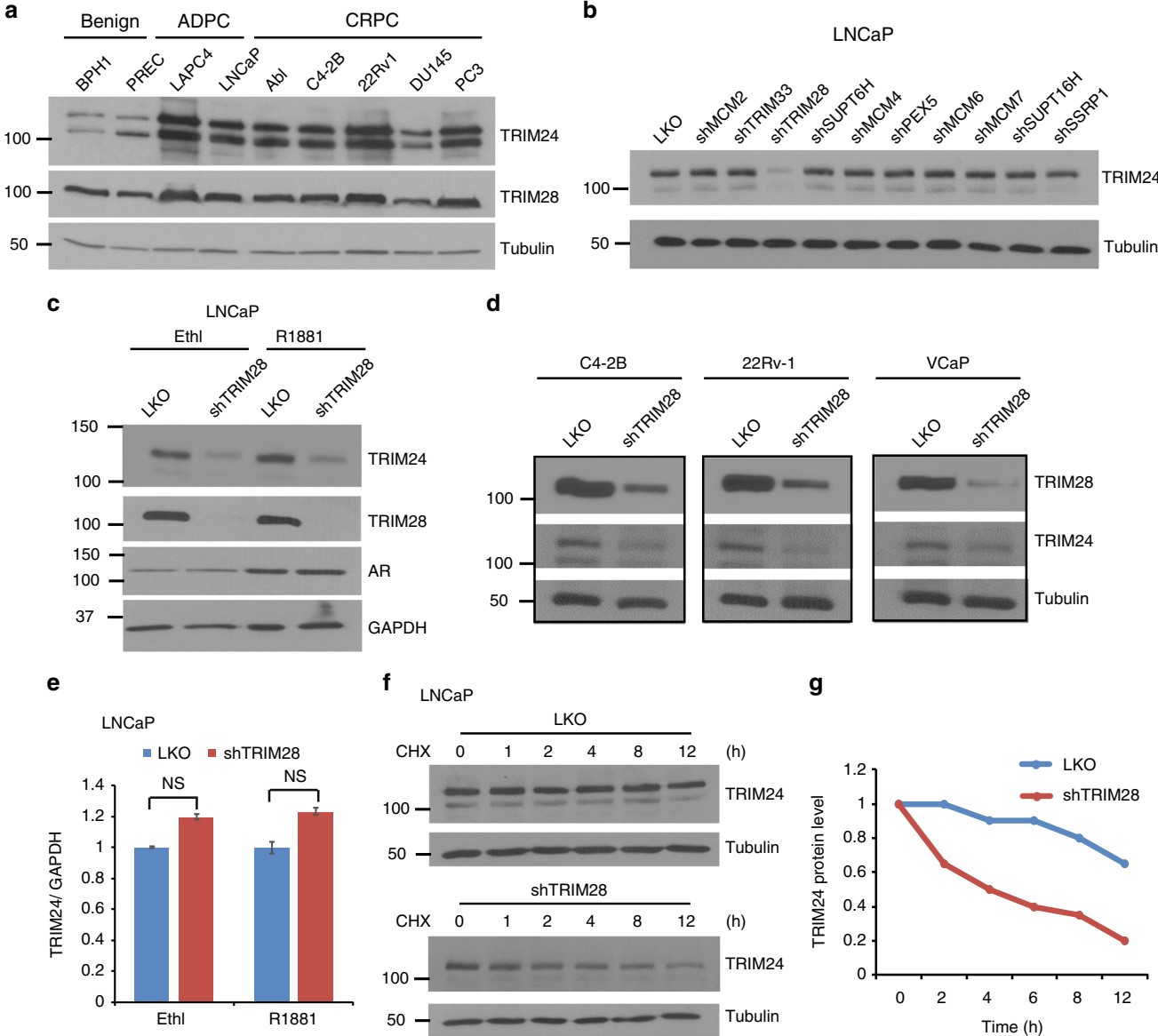

**Fig. 1** TRIM28 is a positive regulator of TRIM24 protein stability. **a** Equal amount of protein lysates from a panel of prostate cells were loaded for SDS-PAGE and western blot analysis was carried out to detect expression of TRIM24 and TRIM28. Tubulin was used as loading control. **b** TRIM28, but not other interactors, positively regulate TRIM24 protein level. Lentiviral supernatant containing pLKO control or shRNAs against top 10 TRIM24-interactors were used to infect LNCaP cells. Four days after infection, protein lysates were harvested and immunoblot performed to examine the expression of TRIM24. GAPDH was used as a loading control. **c** TRIM28 positively regulates TRIM24 protein. LNCaP cells expressing control (pLKO) or shTRIM28 were hormone-starved for 3 days and treated with either ethanol or 1 nM R1881 for 48 h. Protein lysates were harvested and subjected to immunoblotting analysis using indicated antibodies. **d** PCa cell lines expressing control (pLKO) or shTRIM28 were subjected to western blot analysis. Tubulin was used as a loading control. **e** TRIM24 transcript expression is not dependent on TRIM28. RNA was extracted from LNCaP cells expressing control (pLKO) or shTRIM28 and subjected to qRT-PCR analysis. Data shown is mean (± SEM, $n = 3$). NS: not significant. **f**, **g** TRIM28 prolongs the half-life of TRIM24 protein. LNCaP cells infected with shCtrl or shTRIM28 lentiviruses were treated with 100ug/ml Cyclohexamide and protein lysates were collected at the indicated times for western blot analysis (**f**). TRIM24 protein level was quantified and plotted relative to the 0 time point (**g**)

Bacterially expressed and purified GST-tagged TRIM24 mutants coupled to GSH beads were used to pull-down LNCaP protein lysates. First-round of GST pull-down assays showed that TRIM24 N-terminal is required for its interaction with TRIM28 (Fig. 2d). To further narrow down to the interacting domains, we created smaller GST-tagged TRIM24 truncation mutants of the N-terminal region and second-round of GST pull-down pinpointed the B1B2 domain (aa100–260) as essential for TRIM28 interaction (Fig. 2e). In a reciprocal experiment, a series of SFB-tagged TRIM28 truncation mutants were generated

(Fig. 2f) and infected into LNCaP cells along with GST-TRIM24 N-terminal construct (aa1–400). GST-TRIM24-N pull-down assay followed by western blotting with Flag antibody revealed that the entire N-terminal region (aa1–400) of TRIM28, including RING, B1B2, and BBC domains, is required for its interaction with TRIM24 (Fig. 2g). In summary, the B1B2 domain of TRIM24 interacts with the N-terminal region of TRIM28.

Lastly, to determine whether physical interaction with TRIM28 is required for TRIM24 protein stability, we performed a rescue

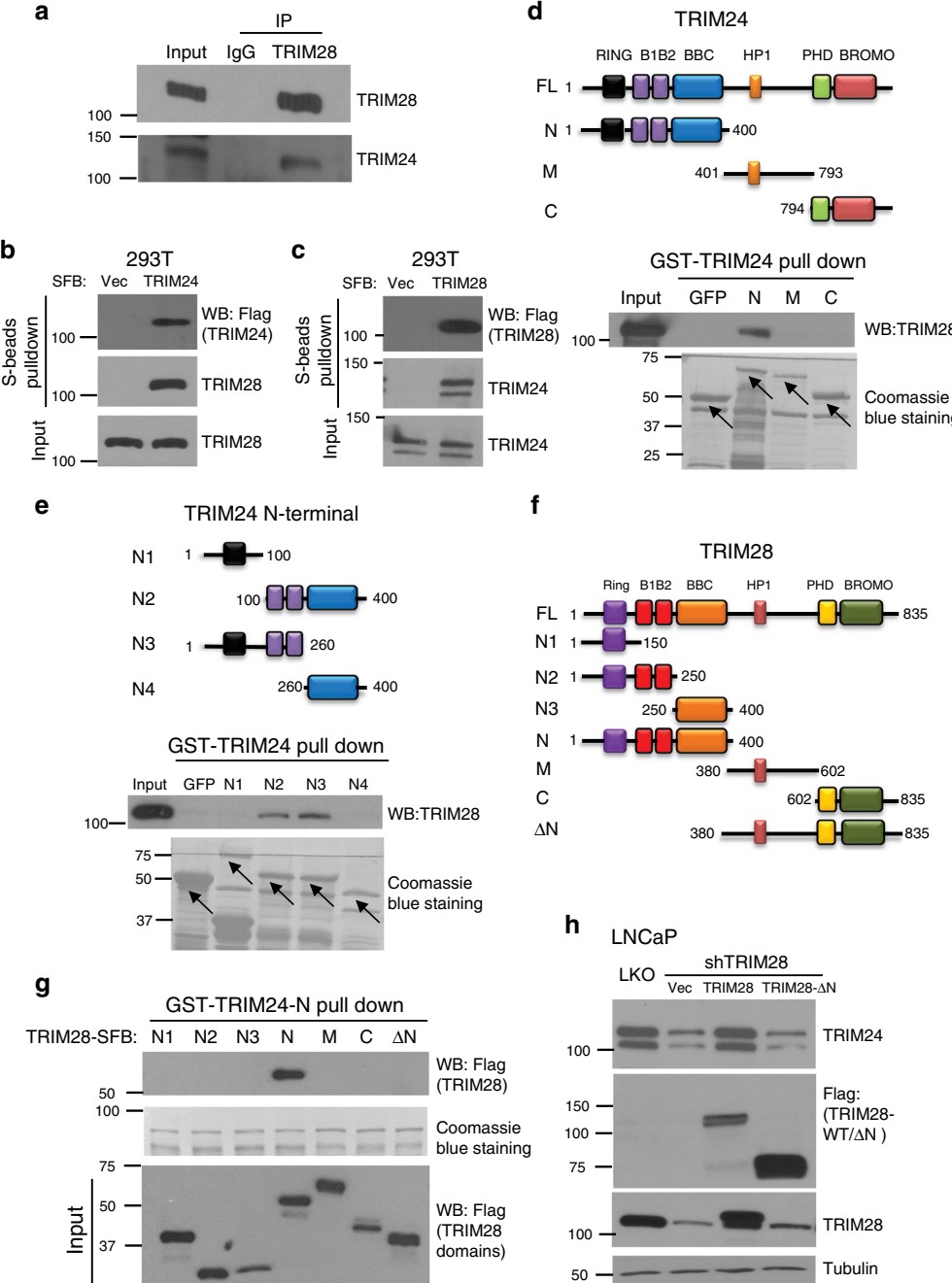

**Fig. 2** Physical interaction with TRIM28 is required for TRIM24 protein stability. **a** TRIM24 interacts with TRIM28 in vivo. LNCaP cell lysates were subjected to immunoprecipitation using a TRIM28 antibody and the eluted protein was analyzed by immunoblotting using anti-TRIM24 and anti-TRIM28 antibodies. **b**, **c** TRIM28 and TRIM24 proteins interact. 293T protein lysates expressing TRIM24-SFB (**b**) or TRIM28-SFB (**c**) was pulled down by S-beads and eluted proteins were analyzed by immunoblotting. **d**, **e** TRIM24 utilizes N-terminal B1B2 domain to interact with TRIM28. Schematic illustration of a series of TRIM24 deletion mutants used in this study (upper panel, **d**, **e**). Bacterially expressed and purified GST-tagged TRIM24 mutants or GFP (negative control) coupled to GSH beads were used to pull-down LNCaP protein lysates. The precipitated protein complexes were subjected to western blot analysis using anti-TRIM28 antibody (lower panel, **d**, **e**). Coomassie blue was used to stain GST-tagged protein coupled to the beads. Arrow depicts intact protein. **f**, **g**. N-terminal region of TRIM28 interacts with TRIM24. Schematic illustration of wild-type and deletion mutants of TRIM28 used in this study (**f**). Bacterial recombinant GST-TRIM24 (1–400aa) protein was incubated with LNCaP cell lysates expressing various SFB-tagged TRIM28 deletion mutants. GST (TRIM2 1–400aa) pull-down were performed using GSH beads and eluted proteins were analyzed with use of anti-flag antibody to detect SFB-tagged TRIM28 deletion mutants (**g**). **h** Physical interaction with TRIm28 is required for TRIM24 stabilization. LNCaP cells expressing pLKO, shTRIM28, co-expressing shTRIM28 + TRIM28 wild-type or shTRIM28 + TRIM28-ΔN-SFB (TRIM24 binding-deficient mutant) were harvested and protein lysates was subjected to immunoblotting using indicated antibodies

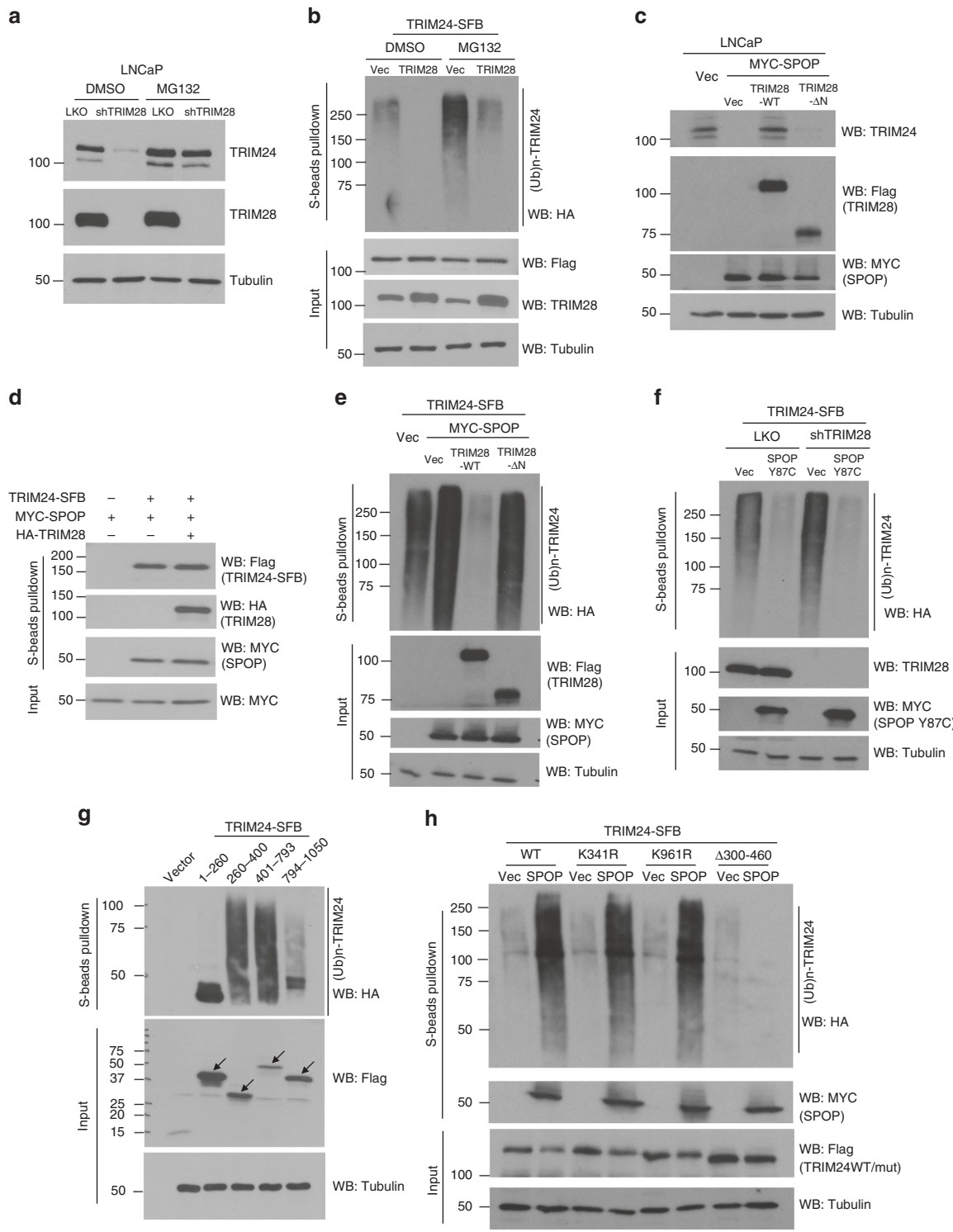

experiment taking advantage of an N-terminal deletion mutant (TRIM28-ΔN) that is unable to interact with TRIM24. LNCaP cells were subjected to TRIM28 knockdown followed by concomitant re-expression of control, wild type, or ΔN TRIM28 constructs. Western blot analysis indicated that wild type, but not TRIM28-ΔN, restored TRIM24 protein that were lost upon TRIM28 knockdown (Fig. 2h). In aggregates, these data support that TRIM28 protein interacts with TRIM24 protein to enhance its stability.

**TRIM28 protects TRIM24 from SPOP-mediated degradation.** To determine how TRIM28 stabilizes TRIM24 protein, we treated control or TRIM28-knockdown LNCaP cells with DMSO or proteasome inhibitor MG132. Western blot analysis showed that MG132 blocked TRIM24 protein degradation in TRIM28-knockdown cells (Fig. 3a), suggesting an ubiquitin-proteasome-mediated pathway. To determine whether TRIM28 prevents TRIM24 protein ubiquitination, we overexpressed control and TRIM28 in 293T cells that were co-transfected with TRIM24-SFB

**Fig. 3** TRIM28 protects TRIM24 from SPOP-mediated ubiquitination and degradation. **a** Loss of TRIM28 causes proteasome-mediated degradation of TRIM24. LNCaP cells expressing shCtrl or shTRIM28 were treated with DMSO or 10 uM MG132 for 8 h and protein lysates were subjected to western blot analysis using indicated antibodies. **b** Overexpression of TRIM28 abolishes ubiquitination of TRIM24. HEK293T cells transiently overexpressing TRIM24-SFB, HA-Ub and empty vector or pCDNA-TRIM28 were treated with DMSO or 20uM MG132 for 4 h and then harvested. S-beads were used to pull-down TRIM24-SFB from cell extract and bound proteins were analyzed by immunoblotting using indicated antibodies. **c** TRIM28 abolishes SPOP-mediated degradation of TRIM24 protein. LNCaP cells with transient overexpression of either empty vector or Myc-SPOP with co-expression of vector, TRIM28-SFB or TRIM28-ΔN-SFB were harvested and subject to immunoblot analysis. **d** TRIM28 does not affect SPOP binding to TRIM24. HEK293T cells co-transfected with TRIM24-SFB and empty vector, Myc-SPOP or Myc-SPOP and HA-TRIM28 were treated with 20uM MG132 for 24 h and then harvested. S-beads were used to pull-down TRIM24-SFB from cell extract, bound proteins were analyzed by immunoblotting using indicated antibodies. **e** TRIM28 overexpression diminishes SPOP-mediated ubiquitination of TRIM24. S-beads pull-down was performed using protein extracts of MG132-treated HEK293T cells co-expressing TRIM24-SFB and HA-Ub and empty vector, MYC-SPOP, MYC-SPOP and Flag-TRIM28 or MYC-SPOP and Flag-TRIM28-ΔN. **f** TRIM28 knockdown enables SPOP-mediated ubiquitination of TRIM24. LNCaP cells were transfected with TRIM24-SFB, HA-Ub, and various constructs as indicated. After 4 h of 20 uM MG132 treatment, cells were harvested and cell lysates were subjected to S-beads pull-down followed by immunoblotting. **g** Various TRIM24-SFB deletion constructs were co-transfected with HA-Ub. Cells were treated with 20 μM MG132 treatment for 4 h and S-beads was used to pull-down TRIM24-SFB fragments. Ubiquitinated TRIM24 was detected by anti-HA antibody. **h** Ubiquitination of TRIM24 by SPOP occurs between 300–460aa. Wild type, K341R, K961R, and Δ300–460 mutants of TRIM24-SFB were co-transfected with HA-Ub with or without Myc-SPOP. Cells were treated with 20 μM MG132 treatment for 24 h and S-beads was used to pull-down TRIM24-SFB wild type or mutants. Ubiquitinated TRIM24 was detected by anti-HA antibody

and HA-Ubiquitin (Ub). S-beads pull-down of TRIM24-SFB followed by western blotting by HA (-Ub) antibody revealed a remarkable decrease of ubiquitinated TRIM24 in cells with TRIM28 overexpression (Fig. 3b). To preclude the possibility that such decrease is due to faster degradation of ubiquitinated TRIM24, we treated the cells with MG132 to block degradation and again found that TRIM28 overexpression dramatically reduced the levels of ubiquitinated TRIM24, indicating a role in suppressing active ubiquitination of TRIM24.

As TRIM24 is a substrate of SPOP[10,19], we hypothesized that TRIM28, through interacting with TRIM24 protein, might disrupt SPOP-mediated TRIM24 binding, ubiquitination, and/ or degradation. We first overexpressed SPOP in LNCaP cells and indeed observed a massive loss of TRIM24 protein (Fig. 3c). Remarkably, overexpression of TRIM28 in these cells fully restored TRIM24 protein levels, while TRIM28-ΔN mutant, which is incapable of interacting with TRIM24 protein, failed to rescue TRIM24 expression (Fig. 3c). Further, we found that TRIM24 protein, which was dramatically increased upon SPOP depletion as expected, was no longer dependent on TRIM28 expression in the absence of SPOP (Supplementary Fig. 3a). Taken together, these results strongly support that TRIM28 stabilizes TRIM24 through blocking its degradation by SPOP.

Next, to investigate whether this is due to a disruption of SPOP binding to TRIM24, we performed S-beads pull-down of TRIM24 in 293T cells co-transfected with TRIM24-SFB, Myc-SPOP, and/ or HA-TRIM28. Western blot analysis confirmed a strong interaction between TRIM24 and SPOP as expected (Fig. 3d). However, this interaction was not impaired by concurrent overexpression of TRIM28 in these cells, suggesting that TRIM28 is unable to disrupt TRIM24-SPOP interaction. Analogous experiments demonstrated that TRIM28 binding to TRIM24 also did not affect the ability of TRIM24-SPOP to recruit other E3 ligase complex proteins such as CUL3 and RBX1, which target substrate ubiquitination (Supplementary Fig. 3b, c). We next hypothesized that TRIM28, albeit unable to disrupt TRIM24 protein interaction with the E3 ligase complex, might undermine its ability to ubiquitinate TRIM24. To test this, we performed S-beads pull-down of TRIM24-SFB in 293T cells that were co-transfected with TRIM24-SFB and HA-Ubiquitin. Western blot analysis confirmed that SPOP overexpression led to a marked increase of TRIM24 ubiquitination (Fig. 3e). Importantly, over-expression of wild-type TRIM28 in these cells fully abolished TRIM24 ubiquitination by both exogenous and endogenous

SPOP, while TRIM28-ΔN mutant failed to rescue. Conversely, TRIM28 knockdown greatly increased TRIM24 ubiquitination, but only in the presence of SPOP, suggesting a blockade of SPOP function (Supplementary Fig. 3d). It was previously reported that PCa-associated SPOP mutants impair TRIM24 ubiquitination[10]. Consistent to this finding, expression of representative SPOP mutants Y87C (Fig. 3f), W131G, and F133V (Supplementary Fig. 3e) fully abolished ectopically expressed TRIM24 ubiquitination, supporting a dominant-negative role over endogenous wild-type SPOP as previously reported[10]. Importantly, while TRIM28 knockdown restored TRIM24 ubiquitination in the control cells, it did not have effects in the presence of SPOP mutant. This data suggest that TRIM28 stabilizes TRIM24 through blocking TRIM24 ubiquitination by SPOP.

To determine whether TRIM24 ubiquitination sites are adjacent to TRIM28-bound domain (aa100–260), we expressed SFB-TRIM24 domain constructs along with ubiquitin in 293T cells. S-beads pull-down followed by HA-ubiquitin western blotting revealed that aa260–400 and aa401–793 truncated proteins were strongly ubiquitinated, while aa1–260 and aa794–1050 was not ubiquitinated (Fig. 3g). To further narrow down sites of TRIM24 ubiquitination, we examined post-translational modification sites on TRIM24 (www.phosphosite. org) and identified five ubiquitination sites: K303, K325, K341, K458, and K1002 (Supplementary Fig. 3f). Out of these, K303/ K325/K341/K458 have previously been detected in 293T cells using single-step immunoenrichment of fractionated ubiquiti-nated peptides and mass spectrometry[31,32]. In particular, K341 was reported to be sensitive to MG132 treatment and predicted to be involved in proteasomal degradation. We also performed mass spectrometry but only detected K961 ubiquitination (Supple-mentary Fig. 3g), probably due to technical challenges associated with ubiquitinated protein fragmentation. To validate these candidate sites, we constructed K341R, K961R, and a TRIM24 mutant with deletion from 300–460aa covering K303/K325/K341/ K458. Ubiquitination assays revealed that TRIM24 Δ300–460 mutant was no longer ubiquitinated by SPOP, while K341R and K961R remained ubiquitinated, suggesting that SPOP likely ubiquitinates multiple lysine sites within 300–460aa region (Fig. 3h). Further, it is known that nearby lysines can become targeted when the major ubiquitination site is mutated[33]. Interestingly, Δ300–460 mutant retained the ability to bind SPOP, analogous to TRIM28, which inhibits TRIM24 ubiquitina-tion without blocking its interaction with SPOP (Fig. 3h). These data support that TRIM28 binding to aa100–260 of TRIM24

prevents SPOP-mediated ubiquitination of TRIM24 at nearby lysine residues within aa300–460 (Supplementary Fig. 3h).

**TRIM28 facilitates TRIM24 transcriptional program**. We sought to examine whether TRIM28 and TRIM24 co-occupies on the chromatin. As previous studies have reported TRIM24 and TRIM28 as coactivators of AR[17,19,26], we analyzed their cistrome in LNCaP cells cultured in the presence of androgen. Since TRIM24 antibodies did not work well in our ChIP-seq experiments, we generated LNCaP cells stably expressing HA-TRIM24 (Supplementary Fig. 4a) and performed HA (TRIM24) ChIP-seq and TRIM28 ChIP-seq in parallel. ChIP-seq analysis demonstrated that nearly 60% of TRIM24-bound genomic loci were also occupied by TRIM28, while the latter had much more abundant occupancy across the genome (Fig. 4a, b). Motif analysis revealed that the most enriched motifs within the TRIM24 binding sites were the forkhead motif followed by ARE (androgen response elements) (Supplementary Data 1). This is in good agreement with a recent report that TRIM24 augments AR signaling and co-occupies active enhancers that are marked by H3K27ac[19], which also strongly overlap with FOXA1[34]. As TRIM28 had nearly 30,000 binding sites that were not co-occupied by TRIM24, to gain some insights on the functionality of these binding events we examined these regions and found that CTCF was the most represented transcription factor binding motif and forkhead motif the second most enriched. TRIM28 is known to colocalize with many DNA damage response factors at sites of DNA double-strand breaks[22]. A recent study has reported that chromosome loop anchors bound by CTCF and cohesion are vulnerable to DNA double-strand breaks[35], which may explain the enrichment of CTCF within our TRIM28 binding sites. Analysis of the distribution of TRIM24 and TRIM28 binding sites relative to their nearest genes revealed strong TRIM24 binding at promoters (Fig. 4c), being consistent with the previous reports[19]. In contrast, TRIM28 was much more abundant at enhancers including intragenic and intergenic regions. In addition, we observed strong TRIM28 occupancy at the promoters and 3′ coding exons of zinc finger (ZNF) genes (Supplementary Fig. 4b), in good agreement with previous analyses of TRIM28 cistrome in 293 cells[36,37].

To further understand the downstream genes/pathways of TRIM28 in contrast to TRIM24 in LNCaP cells with active androgen stimulation, we performed gene expression analysis of these cells subjected to control, TRIM24, or TRIM28 knockdown. We then utilized gene ontology analysis to compare TRIM24- or TRIM28-regulated genes with curated HALLMARK gene sets present in the Molecular Signatures Database (MSigDB)[38]. Among the top ten most enriched biological processes, six of them were common between TRIM24- and TRIM28-activated gene sets (Fig. 4d). These include androgen response, G2M checkpoints, and E2F targets, which were previously shown to be regulated by TRIM24[19]. Further, we found that DNA repair, spermatogenesis, Myc targets, and MTORC1 signaling genes were uniquely regulated by TRIM28. These results are consistent with reported roles of TRIM28 during these biological processes[22,39–41]. On the other hand, TRIM28-repressed gene targets enriched for biological processes such as apoptosis, inflammatory response, and TNFα signaling (Supplementary Fig. 4c).

As TRIM28 stabilizes TRIM24 protein, we asked whether TRIM28 enhances TRIM24 chromatin occupancy. To address this, we performed chromatin fractionation followed by western blot analysis and observed that chromatin-bound TRIM24 was indeed diminished upon TRIM28 knockdown in PCa cells (Fig. 4e). We subjected LNCaP cells stably expressing HA-TRIM24 to control or TRIM28 knockdown and indeed observed a reduced amount of HA-TRIM24 (Supplementary Fig. 4d). HA

(TRIM24) ChIP followed by qPCR analysis of several known TRIM24-target genes[19] revealed greatly reduced HA-TRIM24 occupancy following TRIM28 knockdown (Fig. 4f). Next, to examine how TRIM28 regulates TRIM24 transcriptional program, we performed triplicate microarray experiments profiling LNCaP cells maintained in regular medium with pLKO control, TRIM24, TRIM28 knockdown, or TRIM28 knockdown with TRIM24 rescue. Clustering analysis followed by Treeview revealed that more than half of the genes that were decreased upon TRIM24 depletion (i.e. TRIM24-induced genes) were also downregulated by TRIM28 knockdown and a significant portion of these genes could be restored by concurrent replenishment of TRIM24 (Fig. 4g). Consistently, gene set enrichment analysis (GSEA) demonstrated that TRIM24-induced genes were significantly downregulated following TRIM28 knockdown and rescued by ectopic TRIM24 expression in TRIM28-depleted cells (Supplementary Fig. 4e-f). On the other hand, nearly all genes that were upregulated upon TRIM24 depletion (i.e. TRIM24-repressed gene) were also increased by TRIM28 knockdown, suggesting a strong role of TRIM28 in gene repression, being consistent with previous reports[42,43]. Importantly, a vast majority of these genes could be rescued by TRIM24 replenishment in TRIM28-depleted cells (Fig. 4h). Lastly, qRT-PCR analysis of known TRIM24-induced genes validated that they were decreased by TRIM24 or TRIM28 knockdown but restored upon TRIM24 replenishment, supporting that TRIM28 positively regulates TRIM24 transcriptional program (Fig. 4i).

**TRIM28 enhances AR signaling in prostate cancer**. As recent studies have shown TRIM28 is an AR coactivator that induces AR activities in reporter assays[26], we asked whether TRIM28 enhances AR transcriptional program globally. To test this, we utilized ChIP-seq to map AR cistrome in LNCaP cells with control, TRIM24, or TRIM28 knockdown (Fig. 5a). First, we noted that the depletion of TRIM24 or TRIM28 did not significantly alter AR protein levels (Fig. 5a). ChIP-seq analysis revealed a remarkable reduction of AR binding events in TRIM28-depleted cells when compared to control LNCaP cells, whereas AR binding in TRIM24-knockdown cells primarily showed a shift to new sites, indicating that TRIM28 is required for AR chromatin binding while TRIM24 may play a reprogramming role (Fig. 5b). For this study, we focused on AR binding events that pre-exist in LNCaP cells and found that both TRIM24 and TRIM28 knockdown greatly reduced their ChIP-seq enrichment intensity (Fig. 5c, d). For example, UCSC genome browser view of ChIP-seq tracks demonstrated striking decreases of AR binding at the enhancers of prototype AR-target genes KLK3, TMPRSS2, and KLK2 following TRIM24 or TRIM28 knockdown (Fig. 5e). The data was further validated by ChIP followed by qPCR using site-specific primers (Fig. 5f).

We next asked whether this could be translated into transcriptional regulation of target gene expression. First, using a luciferase reporter assay, we demonstrated that knockdown of TRIM24 or TRIM28 greatly reduced PSA promoter-enhancer transcriptional activities (Fig. 5g). Further, GSEA analysis revealed that androgen-induced genes are significantly downregulated in TRIM28- or TRIM24- depleted cells, while androgen-repressed genes, on the other hand, are restored, supporting TRIM28 and TRIM24 as coactivators of AR (Fig. 5h & Supplementary 5a–b). Lastly, qRT-PCR analysis confirmed that knockdown of TRIM28 or TRIM24 greatly reduced the transcription of AR-induced genes PSA, TMPRSS2, and KLK2 in LNCaP as well as C4-2B cells (Fig. 5i & Supplementary Fig. 5c). In aggregates, these results suggest that, like TRIM24, TRIM28 is also an important activator

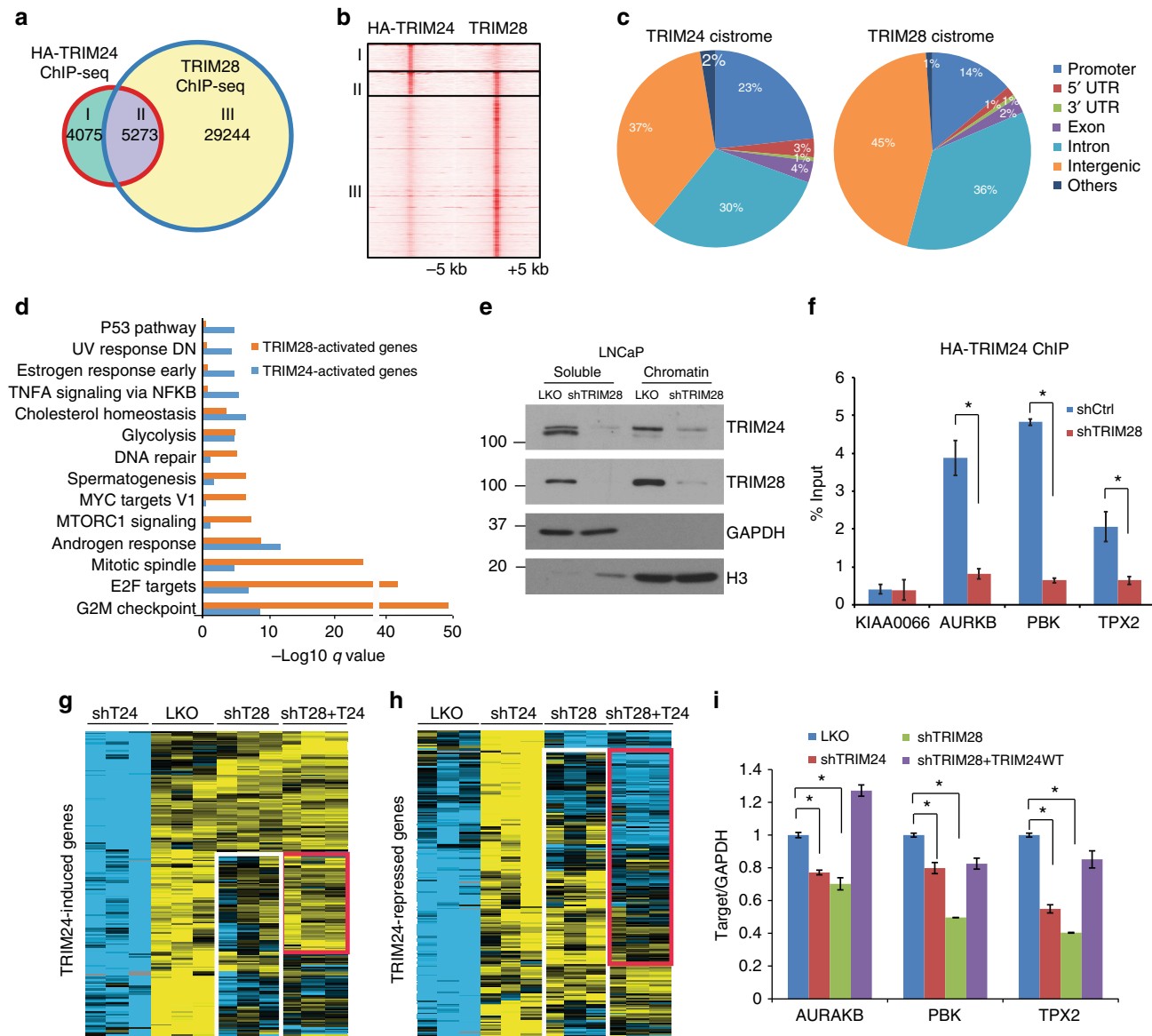

**Fig. 4** TRIM28 enhances TRIM24 transcriptional program. **a**, **b** TRIM28 co-occupies TRIM24 binding sites. LNCaP cells stably expressing HA-tagged TRIM24 were hormone-starved for 3 days, stimulated with 10 nM R1881 for 16 h, and then subjected to TRIM28 and HA (TRIM24) ChIP-seq. Venn diagram (**a**) and Heatmap (**b**) showing overlapping of TRIM28 and TRIM24 binding sites. **c** Genomic distribution of the HA-TRIM24 and TRIM28-specific cistromes. Genomic elements were defined as promoters (2 kb upstream to 2 kb downstream of a TSS—transcription start site), 5′UTR, 3′UTR, coding exons, introns, distal intergenic regions, and others. **d** HALLMARK gene sets enriched by TRIM28- or TRIM24-activated genes. TRIM24- or TRIM28-activated genes were determined by microarray profiling of LNCaP cells expressing pLKO, shTRIM24 or shTRIM28 in the presence of R1881 and subsequently subjected to GSEA analysis. The FDR $q$-values of enriched GO terms are indicated on the x-axis. **e** TRIM28 stabilizes chromatin-associated TRIM24. Subcellular fractionation was carried out in control or TRIM28 knockdown LNCaP cells. The protein lysate from soluble (Sol) and chromatin (Chr) fraction was subjected to immunoblotting. **f** Knockdown of TRIM28 reduces TRIM24 genomic occupancy. LNCaP cells stably expressing HA-TRIM24 were transduced with pLKO or shTRIM28 containing lentivirus. HA-ChIP were then performed and ChIP-enriched ed DNA was subjected to qRT-PCR analysis to detect HA-TRIM24 enrichment at selected TRIM24-induced genes. Data shown is mean (± SEM, $n = 3$). *$P < 0.05$ by Student's $t$-test. **g–i** TRIM28 co-regulates TRIM24 transcription program. Gene expression in LNCaP cells expressing pLKO, shTRIM24, shTRIM28, or shTRIM28 and ectopic TRIM24 re-expression were profiled in triplicate experiments. A total of 220 TRIM24-induced genes (**g**) with adjusted $p$-values <0.05 and 4-fold changes and 215 TRIM24-repressed (**h**) genes with adjusted $p$-values <0.05 and 3.5-fold changes were identified and clustered across all samples. TRIM24-regulated genes that are similarly regulated by TRIM28 are highlighted by white box and the subsets of those that are rescued by TRIM24 re-expression are highlighted by red box. Several known TRIM24-target genes were analyzed by qRT-PCR (**i**). Data shown is mean (± SEM, $n = 3$). *$P < 0.05$ by Student's $t$-test

of AR chromatin binding and transcriptional regulation of target genes.

**TRIM28 is upregulated during PCa progression**. To determine the clinical relevance of TRIM28 regulation of TRIM24 and AR

program, we first examined TRIM28 expression in previously published microarray datasets profiling gene expression in human PCa tissues. We observed highly significant upregulation of TRIM28 mRNA in metastatic PCa as compared to benign and/or localized PCa in multiple microarray datasets[3,44,45] (Fig. 6a, b). In

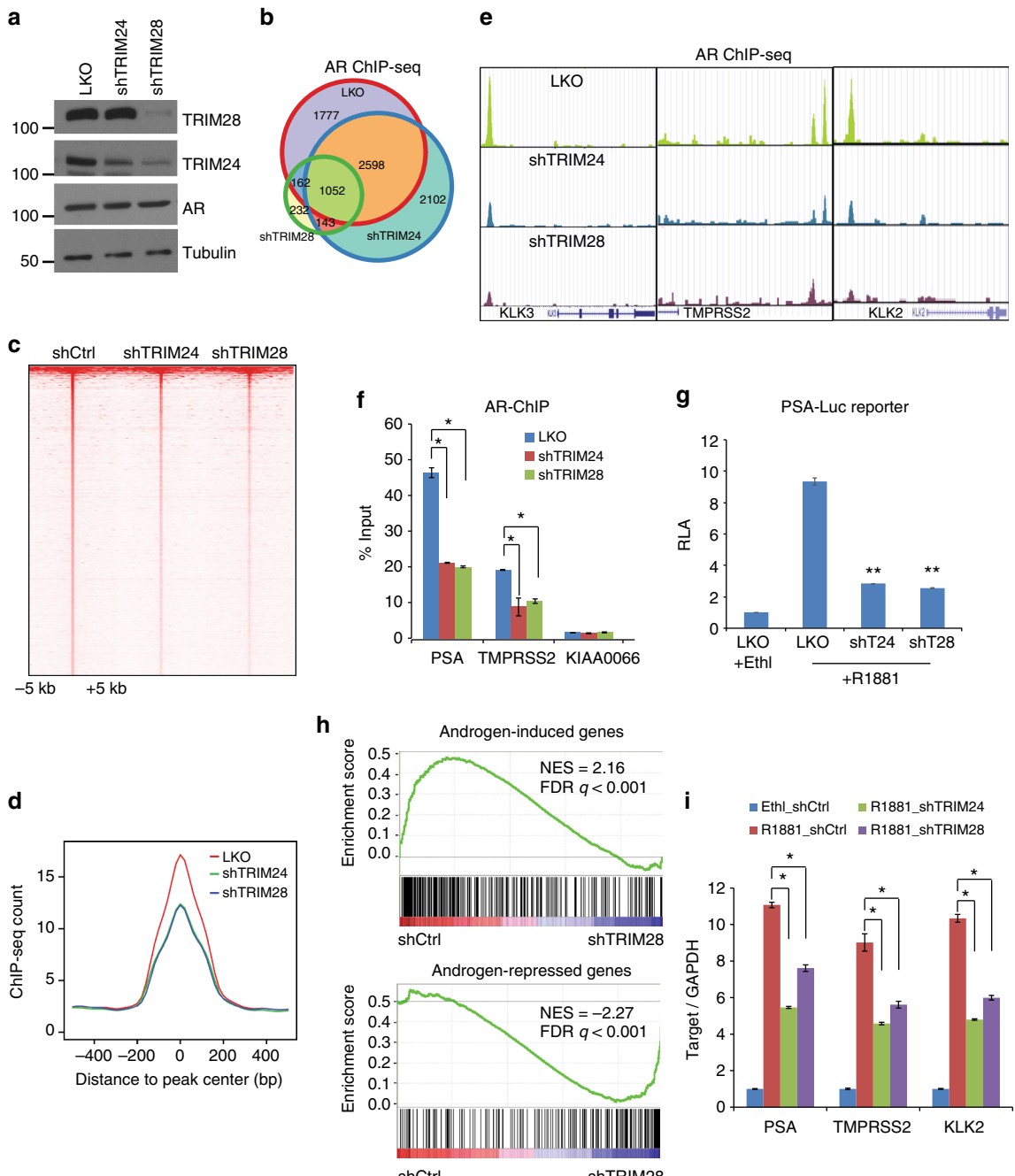

**Fig. 5** TRIM28 enhances AR signaling in prostate cancer. **a** Western blot analysis of protein lysates of LNCaP cells treated with pLKO, shTRIM24, or shTRIM28 with indicated antibodies. **b–e** AR ChIP-seq was performed in control, TRIM24-, or TRIM28-knockdown LNCaP cells. Venn diagram (**b**) shows the overlap of AR binding sites in these cells, where heatmap (**c**) and intensity plots (**d**) depict AR ChIP-seq read intensity across all three samples centered (± 5 kb) at AR binding sites found in control LNCaP cells. AR ChIP-seq signals at several known AR-target genes are shown using UCSC Genome Browser Tracks (**e**). **f** ChIP-qPCR showing AR enrichment at the PSA and TMPRSS2 enhancers. KIAA0066 was used as negative control. Data shown is mean (± SEM, $n = 3$). *$P < 0.05$ by Student's $t$-test. **g** TRIM28 is required for AR transactivation activity. A luciferase reporter construct consisted of PSA enhancer and promoter elements were transfected into pLKO, shTRIM24, and shTRIM28 LNCaP cells. Relative luciferase activities are presented as mean (± SEM, $n = 3$). **$P < 0.01$ by Student's $t$-test. **h** TRIM28 positively regulates AR-mediated gene expression program. GSEA was performed to determine the enrichment of AR-induced (upper panel) or -repressed gene set (lower panel) in gene expression dataset profiling control and TRIM28-knockdown LNCaP cells grown in the presence of androgen. **i** TRIM28 is required for AR-regulated gene expression. LNCaP cells with pLKO, shTRIM24, and shTRIM28 were hormone-starved for 3 days, treated with 1 nM R1881 for 48 h, and subjected to qRT-PCR analysis in comparison to a control LNCaP cells treated with ethanol (Ethl). Data were normalized to GAPDH. Data shown is mean (± SEM, $n = 3$). *$P < 0.05$ by Student's $t$-test

multiple PCa datasets with clinical information[46,47], we found that patients with high levels of TRIM28 showed significantly reduced chances of overall survival, suggesting a prognostic potential of TRIM28 (Supplementary Fig. 6a, b).

Next, to examine TRIM28 protein levels in primary patient tissues and to evaluate its correlation with TRIM24, we performed immunohistochemistry (IHC) in human PCa tumor progression tissue microarrays (TMA). We found that both TRIM24 and

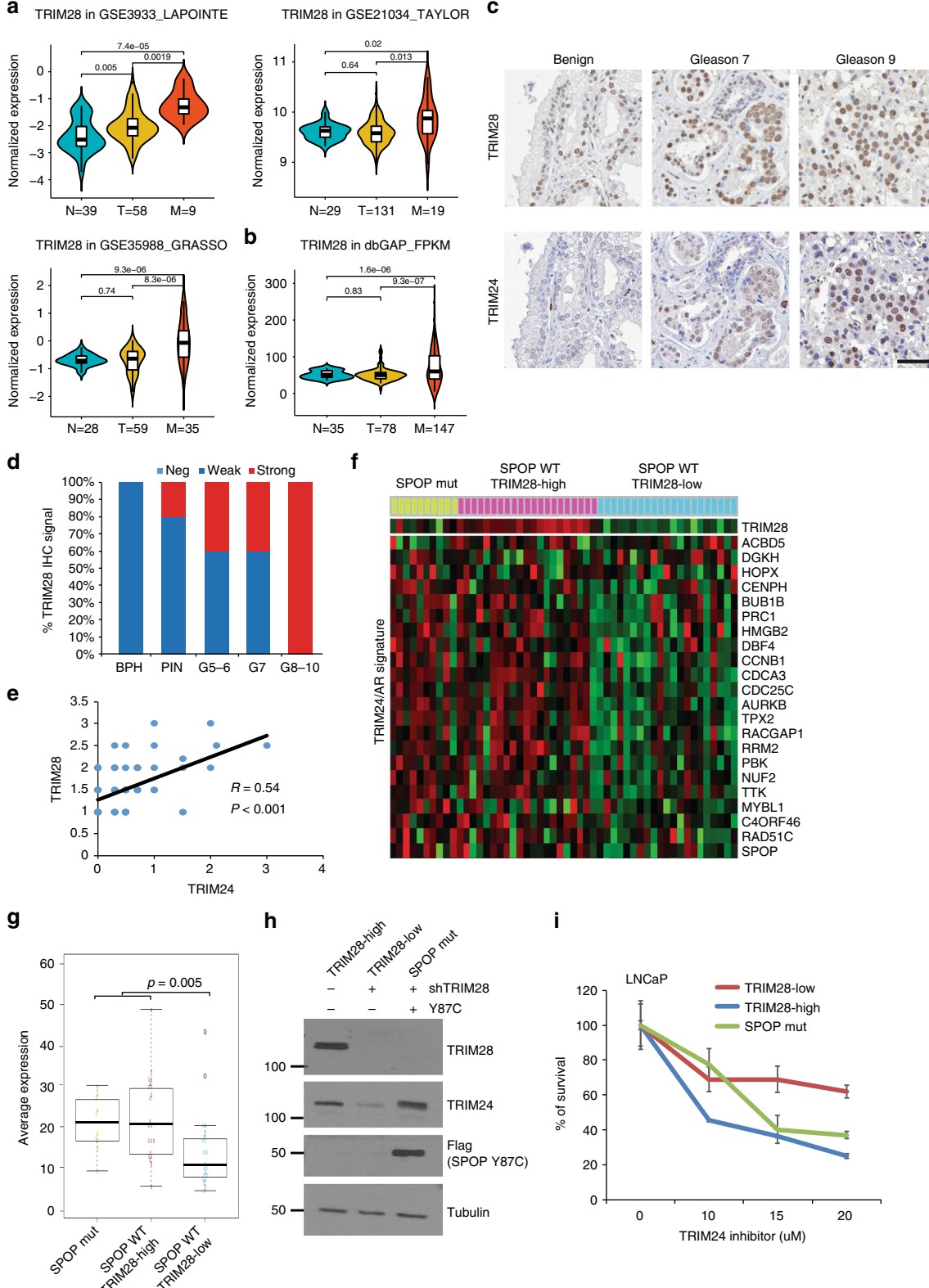

TRIM28 exhibit strong nuclear staining, being consistent with their reported nuclear localization[19,37]. In addition, nuclear staining intensities of TRIM28 and TRIM24 increased from benign to low Gleason tumors, and further to high Gleason tumors (Fig. 6c, d). Moreover, quantification of immunostaining intensities on matched cases showed a positive correlation between TRIM24 and TRIM28 protein levels,

supporting the role of TRIM28 in stabilizing TRIM24 protein (Fig. 6e).

Next, we attempted to determine the relevance of TRIM28 upregulation, similar as SPOP mutations, in enhancing TRIM24 and AR signaling in human PCa. We obtained from cBioportal the SU2C dataset of 150 CRPC samples[4], 33 out of which have missing values and were excluded from subsequent analyses.

**Fig. 6** TRIM28 is upregulated during PCa progression. **a, b** TRIM28 expression levels in benign adjacent (N), localized PCa (T), and metastatic CRPC (M) in publically available prostate cancer microarray profiling datasets (GSE35988, GSE3933, GSE21034) (**a**). RNA-seq data (**b**) from three dbGAP PCa dataset[4,55](phs000915.v1.p1, phs000909.v1.p1, and phs000443.v1.p1) were compiled based on FPKM (fragments per kilobase per million reads). Elements in Violin boxplot: horizontal black line = the median, inside box = interquartile range, whiskers = 95% confidence interval, shape of the violin = frequencies (density plot width) of values, upper and lower needles = the maximum and minimum values. **c–e** TRIM28 expression is elevated in aggressive PCa. Human PCa progression tissue microarray were used for immunohistochemistry staining. Representative staining images were shown (**c**). TRIM28 staining by Gleason Scores was summarized by histogram (**d**) and its correlation with TRIM24 staining determined in (**e**) using Pearson method (test for association/correlation between paired samples). Scale bar = 50 μm. **f, g** TRIM24/AR coactivated genes induced in PCa with either SPOP mutation or TRIM28 elevation. SU2C RNA-seq dataset of 150 CRPC samples[4] were obtained from cBioportal. Exome-sequencing data of each sample were examined to identify the subset with SPOP mutations (SPOP mut, n = 10). The rest of the samples that are SPOP wild type (WT) were further stratified based on their RNA-seq data into TRIM28-High (ranked top 20% in TRIM28 expression, n = 21) and TRIM28-Low (ranked bottom 20%, n = 21). Heatmap (**f**) shows clustered TRIM24/AR coactivated gene expression in these 3 categories, while boxplots (**g**) depicts average expression of these genes. Statistical difference was assessed by Student's *T*-test. Elements in boxplot: Upper whisker = max value excluding outliers, upper bound of the box = 3rd quartile, central line = median, lower bound of the box = 1st quartile, lower whisker = least value excluding outliers. **h, i** PCa with SPOP mutation or TRIM28 elevation are sensitive to TRIM24 inhibitors. LNCaP cells (SPOP WT) were transduced with lentivirus containing pLKO, shTRIM28, or shTRIM28 along with SPOP-Y87C-SFB (**h**). These cells were then treated with increasing doses of TRIM24-bromodomain inhibitor (TRIM24-C34) for 14 days and measured for cell survival in colony formation assays (**i**)

Based on exome-sequencing data, we first divided the 117 samples into SPOP mutations (n = 10) and SPOP wild type (n = 107), the latter of which were further stratified into TRIM28-high (top 20%, n = 21) and TRIM28-low (bottom 20%, n = 21) based on their RNA-seq TRIM28 expression data. Importantly, heatmap analysis revealed that previously reported TRIM24 and AR coactivated genes[19] were in general expressed at much higher levels in PCa subsets with either SPOP mutations or TRIM28 overexpression (Fig. 6f). Boxplot analysis showed that CRPC with either SPOP mutations or TRIM28 overexpression showed a significantly higher average expression of these genes than CRPC with both SPOP wild type and low TRIM28 (Fig. 6g). Further, we observed a trend of lower TRIM28 expression in SPOP-mutant human tumors as well as mouse organoids[48], supporting potentially complementary roles (Supplementary Fig. 6c, d). Consistently, the TRIM24-AR gene signature also showed a strong enrichment for upregulation in TRIM28-high mouse organoids (Supplementary Fig. 6d). Altogether, these data strongly suggest TRIM28 elevation as a new mechanism, other than SPOP mutation, to promote TRIM24 and AR signaling. Two studies have recently developed TRIM24 bromodomain inhibitors, such as TRIM24-C24, with good selectivity over other bromodomain proteins[49,50]. We first confirmed that the TRIM24 inhibitor TRIM24-C34 repressed androgen-induced gene expression at 10 μM range (Supplementary Fig. 6e), being consistent with the notion that TRIM24 is an activator of androgen signaling[19]. Further, we demonstrated that TRIM24-C34 is much more effective in suppressing the growth of AR-positive (LNCaP and C4-2B) than AR-negative (DU145) PCa cells, highlighting the specificity of this compound to target TRIM24-AR axis (Supplementary Fig. 6f).

Since TRIM24 activity is high in PCa with either SPOP mutations or high TRIM28 expression, we argued that these tumors might be sensitive to TRIM24 inhibition. To test this, we attempted to generate isogenic PCa models. As LNCaP cells are SPOP wild type and express high levels of TRIM28 (Fig. 1), they were subjected to either ectopic SPOP Y87C mutant expression or TRIM28 knockdown. Western blot analysis confirmed three isogenic LNCaP lines corresponding to SPOP mutant, TRIM28-high, and TRIM28-low prostate tumors (Fig. 6h). These cells were then treated with increasing doses of TRIM24-bromodomain inhibitor TRIM24-C34. Clonogenic assay revealed that the control TRIM28-high cells and the SPOP-Y87C cells were indeed much more sensitive to TRIM24-C34 inhibitor, with IC50 = 8.6 μM and 12.6 μM respectively, than the TRIM28-low, SPOP-wild-type cells, which had IC50 = 69.7 μM (Fig. 6i). Similar results

were also observed in CRPC cell lines and with additional SPOP mutants W131G and F133V (Supplementary Fig. 6g–h). Taken together, our results support that TRIM28 was upregulated in aggressive human PCa and, like SPOP mutations, enhanced TRIM24 and AR signaling.

**TRIM28 promotes prostate cancer tumorigenesis.** We next attempted to determine the roles of TRIM28 in prostate tumorigenesis. We performed TRIM28 knockdown in androgen-dependent LNCaP and VCaP cells as well as androgen-independent C4-2B and 22Rv1 cells and observed significantly reduced proliferation of (Fig. 7a & Supplementary Fig. 7a). Further, colony formation assay demonstrated that TRIM28 depletion significantly reduced the clonogenic ability of these cells (Fig. 7b & Supplementary Fig. 7b). In a rescue experiment, we illustrated that re-expression of ectopic TRIM24 in TRIM28-knockdown cells (Supplementary Fig. 7c) restored, at least partially, the colony formation ability of PCa cells (Fig. 7c).

To examine the tumorigenic roles of TRIM28 in vivo, we inoculated control, TRIM28-knockdown, or TRIM28-knockdown and TRIM24-re-expressing C4-2B cells into SCID mice and monitored xenograft tumor growth. We found that TRIM28-knockdown cells showed retarded xenograft tumor growth and ectopic re-expression of TRIM24 in these cells partially rescued xenograft tumor growth (Fig. 7d). The differences in tumor growth over time among the three groups were statistically significant (ANOVA, P < 0.001). We dissected out all xenograft tumors at the endpoint (Fig. 7e) and found significantly (T-test, P = 0.001) reduced endpoint tumor weight without affecting overall mice body weight in the TRIM28-knockdown group (Fig. 7f & Supplementary Fig. 7d). Re-expression of TRIM24 showed a trend in rescuing tumor growth but the change was not statistically significant. To verify the manipulation of these proteins at the endpoint, we performed IHC in tumor tissue sections and found that both TRIM28 and TRIM24 protein stained predominantly in the nuclei and their staining intensities were decreased in TRIM28-knockdown tumors (Fig. 7g). In addition, TRIM24 staining, as expected, was rescued in xenograft tumors with ectopic TRIM24 re-expression. Further, KI-67 staining was reduced in TRIM28-knockdown tumors and was partially rescued in TRIM24-reexpressing tumors, being concordant with their respective tumor growth rates. In aggregates, these results suggest prostate tumorigenic role of TRIM28 that is mediated in part by TRIM24.

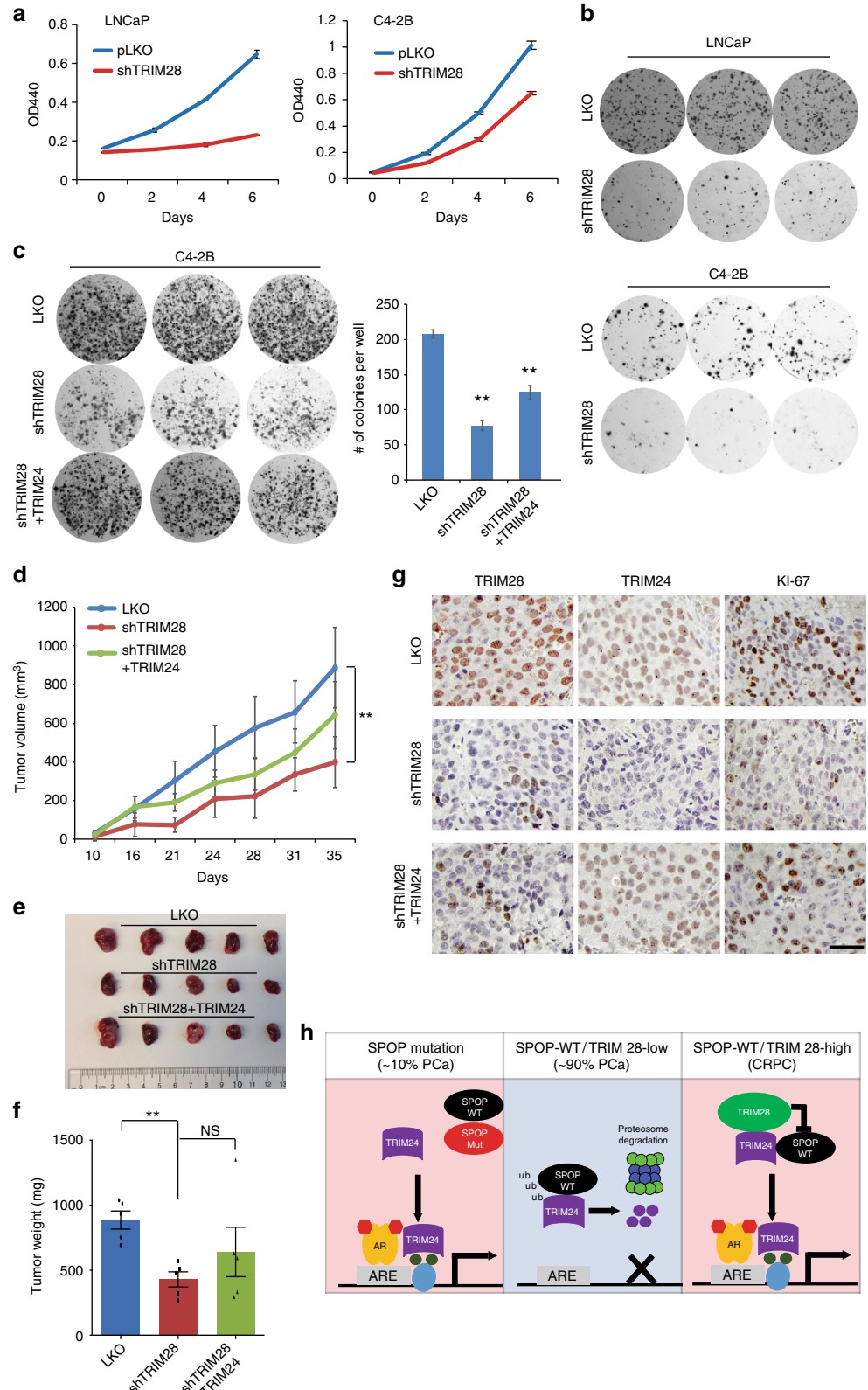

## Discussion

E3 ubiquitin ligase substrate-binding adaptor protein SPOP has been a focused point of investigation in recent years due to its frequent mutations in prostate cancer and endometrial tumors[1,51]. Noteworthy, SPOP mutations are only found in about 12% of primary PCa and only 7% of metastatic CRPC, while

TRIM24 was recently shown broadly upregulated in CRPC[19], suggesting additional regulatory pathways that are particularly important in aggressive PCa. In this study, we identified TRIM28 as a general upstream regulator of TRIM24 protein stability (Fig. 7h). Unlike SPOP mutations, TRIM28 does not disrupt SPOP binding to TRIM24 protein. Rather, TRIM28 expression

**Fig. 7** TRIM28 promotes prostate cancer tumorigenesis in vitro and in vivo. **a**, **b**. TRIM28 is required for prostate cancer growth. Proliferation of control or TRIM28-knockdown LNCaP and C4-2B cells with were evaluated by WST-1 assay (**a**) and colony formation assay (**b**). **c** Colony formation assay was performed using C4-2B cells transduced with lentiviral supernatant of pLKO, shTRIM28, and shTRIM28 along with TRIM24-SFB. After 10–14 days, colonies were stained, imaged, and quantified in a bar graph. Data shown is mean (± SEM, $n = 3$). **$P < 0.01$ by Student's $t$-test. **d**–**g** TRIM28 depletion reduced CRPC xenograft tumor growth in part through decreasing TRIM24. C4-2B cells ($4 \times 10^6$) expressing pLKO, shTRIM28, or shTRIM28 along with TRIM24-SFB were inoculated to the right flanks of CB17-SCID mice. Tumor growth was measured twice per week (**d**) and the difference among groups were determined by ANOVA (**$p < 0.001$). Tumors were excised at the endpoint (**e**) and weighed (**f**). Statistical differences of endpoint tumor weight between experimental and control group was determined by Student's $t$-test. Data shown is mean (± SEM, $n = 5$). *$P < 0.05$; NS: not significant. Tumor tissues from mice were paraffin embedded, sectioned and subjected to immunohistochemistry with the use of anti-TRIM28, TRIM24 and KI-67 antibodies as indicated (**g**). Scale bar = 50 μm. **h** A model depicting broad TRIM24 upregulation by TRIM28 in CRPC. In PCa with SPOP mutations (~ 10% PCa, 7% CRPC), mutant SPOP forms heterodimers with wild-type SPOP to prevent it from binding to and degradation of TRIM24, which subsequently leads to TRIM24 protein increase and enhanced AR transcriptional activities. In a majority of localized PCa that are SPOP wild type and TRIM28 low, SPOP recruits Culin3-RBX1 ubiquitin ligase to TRIM24 for ubiquitination and subsequent degradation in proteasomes. In CRPC wherein TRIM28 is highly elevated, TRIM28 binds to TRIM24 protein to prevent its ubiquitination by SPOP and stabilize TRIM24, thereby enhancing AR signaling and driving CRPC tumor growth

abolishes SPOP-mediated ubiquitination of TRIM24. We showed that TRIM28-TRIM24 interaction is required to prevent TRIM24 ubiquitination by SPOP. TRIM28 binding may conceivably result in conformational changes or the recruitment of cofactors to TRIM24 protein that subsequently interferes or blocks SPOP-mediated ubiquitination. As TRIM28 is highly upregulated in aggressive PCa, this provides a novel mechanism to broad TRIM24 protein elevation in CRPC tumors.

A previous study demonstrated that tumors with SPOP or FOXA1 mutations exhibit the highest AR transcriptional activity among distinct subsets of prostate cancer[2], whereas a recent finding revealed that SPOP mutation is depleted in metastatic CRPC[52], possibly due to selection against androgen-dependent SPOP-mutant cells during androgen deprivation therapy. In contrast, TRIM28 high-tumor cells may be favorably selected as TRIM28 overexpression enhances AR signaling, which is a key mechanism to prostate cancer resistance to androgen deprivation therapy. Being consistent with this, TRIM28 is highly upregulated in CRPC as reported here. Moreover, we found that the genomic background of SPOP mutant and TRIM28-high tumors are also different. SPOP mutant tumors were reported to often harbor CHD1 loss but lack PTEN alteration and ETS gene fusions[1,52]. On the contrary, our analysis revealed a lack of association between TRIM28 level and CHD1, PTEN and ETS alterations (Supplementary Fig. 7e, f). Interestingly, we found that TRIM28-high tumors have significantly higher rates of RB1 loss (Supplementary Fig. 7g, h), which was recently found enriched in metastatic samples[52].

TRIM28 has been previously shown to stabilize a family of KRAB-ZNF proteins through direct interactions, while the underlying mechanism is unclear[53]. In concordance with this, our study showed that TRIM28 interacts with TRIM24 protein to promote its stability and further implicated a role of TRIM28 in blocking SPOP-mediated ubiquitination. In the present study, we used ChIP-seq to identify a large number of TRIM28-bound chromatin regions, which overlapped with more than 50% of TRIM24 binding sites, suggesting TRIM28 as a TRIM24 coactivator. TRIM28 expression increases TRIM24 chromatin binding and transcriptional program, being consistent with the notion that TRIM28 is required for TRIM24 protein stability. We observed stronger degree of dependency of TRIM24-repressed genes on TRIM28 than TRIM24-induced genes, suggesting stronger role of TRIM28 in transcriptional repression as reported in the literature[23,36,42]. Further, like TRIM24, TRIM28 expression is also required for AR chromatin binding. Interestingly, we observed much more profound loss of AR binding events in TRIM28- than TRIM24-depleted ones, which may be in part due to stronger depletion of TRIM24 in the former cells (Fig. 5). However, it is also possible that TRIM28 may directly interact

with AR as previously suggested[26] or it may regulate AR cistrome through other TRIM24-independent mechanisms. Further, the reported role of TRIM28 in heterochromatin formation and gene repression may add additional complexity to its regulation of gene expression programs.

A recent GWAS study of PSA-screened Chinese men has reported PCa risk-associated SNPs within 19q13.4, although the risk allele is more than 4 mb away from the TRIM28 gene[54]. Using luciferase reporter assays TRIM28 has been shown as a coactivator of AR that induces PSA promoter activity[26]. However, TRIM28 cistrome and regulation of gene expression in PCa cells remain to be studied, despite a recently study suggesting TRIM28 enrichment at tumor-specific closed chromatin in prostate cancer cells[25]. As TRIM24-AR signature genes are equally activated in TRIM28-high tumors as in SPOP-mutant PCa, TRIM28 expression, easily monitored at mRNA level, could be used to select CRPC patients that might not have SPOP mutation for TRIM24-targeted therapeutics that was previously proposed[19]. This is supported by our data showing that PCa with either SPOP mutant or high TRIM28 expression is sensitized to TRIM24 inhibitors.

## Methods

**Cell lines and chemical reagents**. LNCaP, 22RV1, and VCaP cell lines were obtained from the American Type Culture Collection (ATCC). C4-2B were kind gifts from Dr. Arul Chinnaiyan. Cells were authenticated, free of mycoplasma, and grown in RPMI supplemented with 10% fetal bovine serum, 1% penicillin and streptomycin. MG132 was from Sigma. TRIM24-C34 compound was from Xcessbio. All antibodies used in this study are listed in Supplementary Table 2.

**Construct and quantitative PCR (qPCR)**. TRIM28, TRIM24, and SPOP constructs were first cloned into pCR8 Gateway compatible entry vector, then transferred into pLenti-SFB, pLVX or pDEST-Myc gateway compatible destination vector by LR clonase (Invitrogen). GST-TRIM24 constructs were generated by subcloning into pGEX4T-1 vector. SPOP Y87C/F133V/W131G mutants were generated by Quick-Change II site-directed mutagenesis kit (Aligent Technology) with MYC-SPOP WT as a template. All the plasmids were verified by sequencing. HA-Ubiquitin was a gift from Edward Yeh (Addgene plasmid # 18712). pKH3-TRIM28 was a gift from Fanxiu Zhu (Addgene plasmid # 45569). pCDNA3-myc3-ROC1 and pcDNA3-myc-CUL3 were a gift from Yue Xiong (Addgene plasmid # 20717 and # 19893). For cDNA synthesis, 250 ng of RNA per sample was used for cDNA synthesis using qscript cDNA synthesis supermix (Quantabio). qRT-PCRs were performed using 2xBullseye EvaGreen qPCR MasterMix (MIDSCI) and StepOne Plus (Applied Biosystems). Primers were designed using primer3 and synthesized by Integrated DNA Technologies (Supplementary Table 3). All the shRNA are purchased from sigma (Supplementary Table 4).

**Tandem affinity purification of SFB-tagged protein complexes**. LNCaP cells stably expressing TRIM24-SFB were lysed in NETN (100 mM NaCl, 20 mM Tris-Cl, pH 8.0, 1 mM EDTA, and 0.5% [vol/vol] NP-40) buffer containing protease inhibitors for 20 min at 4 °C. Crude lysates were subjected to centrifugation at $21,100 \times g$ for 30 min. Supernatants were then incubated with streptavidin-conjugated beads (GE Healthcare) for 4 h at 4 °C. The beads were washed three times with NETN buffer, and bounded proteins were eluted with NETN buffer

containing 2 mg/ml biotin (Sigma-Aldrich) for 1 h twice at 4 °C. The elutes were incubated with S-protein beads (EMD Millipore) overnight at 4 °C. The beads were eluted with SDS sample buffer and subjected to SDS-PAGE. Protein bands were excised and subjected to mass spectrometry analysis.

**Protein binding assay and protein ubiquitination assay.** To probe the interaction between endogenous TRIM28 and TRIM24, LNCaP cell lysates were incubated with 2 ug anti-TRIM28 and anti-rabbit IgG (Santa Cruz) overnight at 4 °C. Dynabeads Protein A (Life Technologies), 25 ul per IP, were added the next day, incubated for 2 h at 4 °C before being eluted with SDS sample buffer. For GST-pull down assay, GST-TRIM24 recombinant proteins were first coupled into Glutathione sepharose (GE healthcare). Then, the beads were incubated with cell lysate expressing TRIM28 full length or truncation mutants for 2–3 h. After extensive washing with NETN buffer, the beads were eluted with SDS sample buffer and subjected to western blot analysis. S-beads pull-down were performed using 30 ul S-protein agarose beads (Millipore) for 3 h at 4 °C. To detect endogenous TRIM24 ubiquitination, LNCaP cells transduced with pLKO or shTRIM28 were treated with 20uM MG132 for 4 h. After lysis with NETN, 2 ug anti-TRIM24 was added to lysate overnight at 4 °C, further incubated with Dynabeads Protein A for 2 h at 4 °C. To detect TRIM24-SFB ubiquitination, S-beads were incubated with MG132-treated 293 T lysates expressing TRIM24-SFB and indicated constructs for 3 h at 4 °C. Bound proteins were eluted from the beads with 1XSDS sample buffer for 10 min at RT and the supernatant was transferred to a new centrifuge tube to boil at 95 °C for further 2 min before SDS-PAGE analysis. Uncropped scans for western blot analysis are shown in Supplementary Figs. 8–10.

**WST-1 cell proliferation assay and colony formation assay.** WST-1 cell proliferation assay (Clontech) and the Dual-Luciferase Reporter Assay System (Promega) was performed according to the manufacturer's instruction. For colony formation assay, 2000–5000 cells per well were seeded on 6-well plate. After 2–3 weeks, the cells were first fixed by 4% paraformaldehyde and then stained with 0.05% crystal violet. The colonies were then imaged and counted for quantification.

**Tissue acquisition and immunohistochemistry.** Tissue microarrays containing PCa specimens containing 10 cases each of benign prostatic hyperplasia (BPH), high-grade prostatic intraepithelial neoplasia (PIN), Gleason score 5–6, 7, and 8–10 localized prostate adenocarcinoma were obtained through Cooperative Human Tissue Network (CHTN) at the University of Virginia. FFPE tissues on glass slides were subjected to deparaffinization and rehydration followed by antigen retrieval with citrate acid at pH6, tissues were permeabilized with 0.5% Triton X-100, peroxidase blocked with 0.3% $H_2O_2$, blocked with avidin and biotin, followed by 2% BSA protein block. Then, slides were incubated with anti-TRIM28 at 1:400, anti-TRIM24 at 1:200 dilution for 2 h at RT, washed 5× with TBST and incubated with biotin-conjugated secondary antibody (1:200) for 15 min at RT. After extensive washing with TBST, slides were incubated with streptavidin-HRP (1:500) for 15 min at RT, wash 3× with TBST and incubated with DAB substrate for 5–10 min at RT until desired color intensity was reached. After that, slides were counterstained with hematoxylin for 1 s, extensively washed with water, dehydrated in ethanol, cleared with Xylene and finally mounted by permount. Images of TMA slides were captured with use of Zeiss UVLSM 510 Meta system, exported to TissueFAXs viewer, while images of mice tumor tissues were taken with use of Olympus BX41. Captured images were modified using Photoshop CS4 (Adobe). Immunostaining was quantified using a score of 0 to 3 for intensities of negative, weak, moderate and strong.

**ChIP, ChIP-seq, and data analysis.** Cultured cells were cross-linked with 1% formaldehyde for 10 min and the cross-linking was quenched by 0.125 M glycine for 5 min at room temperature. Cells were then rinsed with cold 1 × PBS twice. The following steps were performed at 4 °C. Cell pellets were resuspended and incubated in cell lysis buffer (5 mM PIPES, pH 8.0, 85 mM KCl, 0.5% NP40) with protease inhibitor (Roche) for 10 min. Nuclei pellets were spun down at 5000° for 5 min, resuspended in nuclear lysis buffer (50 mM Tris-Cl, pH 8.0, 10 mM EDTA, 1% SDS) and then incubated for another 10 min. Chromatin was sonicated to an average length of 500 bp and then centrifuged at $21,000 \times g$ for 15 min to remove the debris. Supernatants containing chromatin fragments were pre-cleared with protein A agarose beads (Millipore) for 20 min and centrifuged at $500 \times g$ for 5 min. To immunoprecipitate protein/chromatin complexes, the supernatants were first diluted 10 times with IP dilution buffer (0.01% SDS, 1.1% Triton X 100, 1.2 mM EDTA, 16.7 mM Tris-Cl, pH 8.1, 167 mM NaCl) and then incubated with 3–5 ug of antibody overnight, then added 50 ul of protein A agarose beads and incubated for 2 h. Beads were washed twice with 1× dialysis buffer (2 mM EDTA, 50 mM Tris-Cl, pH 8.0) and four times with IP wash buffer (100 mM Tris-Cl, pH 9.0, 500 mM LiCl, 1% NP40, 1% Deoxycholate). The antibody/protein/DNA complexes were eluted with 150 ul IP elution buffer (50 mM NaHCO₃, 1% SDS) twice. To reverse the cross-links, the complexes were incubated in elution buffer + 10 ug RNase A and 0.3 M NaCl at 67 °C for overnight. DNA/proteins were precipitated with ethanol, air-dried and dissolved in 100 ul of TE. Proteins were then digested by proteinase K at 45 °C for 1 h and DNA was purified with QIAGEN PCR column and eluted with 30 ul nuclease-free water. ChIP-seq libraries were prepared according to standard protocols using BioScientific's DNA Sample Kit

(Cat# 514101). Libraries were sequenced using Illumina Hi-Seq platforms. Sequence reads were aligned to the Human Reference Genome (assembly hg19) using Burrows-Wheeler Alignment (BWA) Tool version 0.6.1. ChIP-seq reads were analyzed using the HOMER (Hypergeometric Optimization of Motif EnRichment) suite (http://homer.salk.edu/homer/). The total reads of samples or input were matched to the same size by randomly picking reads. Weighted venn diagrams were created by R package Vennerable. The motif percentage of occurrence was created by R packages: ggplot2, scales and gridExtra.

**Microarray and gene set enrichment analysis (GSEA).** Microarray expression profiling was performed using HumanHT-12 v 4.0 Expression BeadChip (Illumina). Bead-level data were preprocessed and normalized by GenomeStudio. Differentially expressed genes were identified by Bioconductor limma package (cutoff $p < 0.005$). Heatmap view of differentially expressed genes was created by Cluster and Java Treeview. GO term enrichment was analyzed using DAVID and plot was drawn by R package ggplot2. GSEA was performed following the manufacturer's instruction. (http://software.broadinstitute.org/gsea/index.jsp)

**Xenograft tumor growth.** All procedures involving mice were approved by the Institutional Animal Care and Use Committee at Northwestern University and complied with all relevant ethical regulations. 3–4-week-old male CB17.SCID (Charles River Laboratory, Wilmington, MA, USA) were used. To evaluate the role of TRIM28 in tumor formation, $4 \times 10^6$ C4-2B cells stably expressing pLKO, shTRIM28 or co-expressing shTRIM28 and TRIM24 were inoculated by subcutaneous injection into the dorsal flank of the mice. Tumor size was measured twice a week, tumor volumes were estimated using the formula $(\pi/6) (L \times W^2)$, where L = the length of tumor and W = the width, and tumor weight was measured after the mice were euthanized. After endpoint was reached, mice were euthanized, tumors were excised for subsequent analysis.

**Statistical analysis.** Mice were assigned at random for xenograft assay. Statistical significance was determined via two-tailed unpaired Student's $t$-test or ANOVA using Microsoft Excel or Prism software (GraphPad). Collected data are presented as mean ± standard error. Significance values were set at *$P < 0.05$, **$P < 0.01$, ***$P < 0.001$.

## Data availability
The authors declare that all the data supporting the findings of this study are available are available from the article and Supplementary Information files, or from the corresponding author upon reasonable request. All high-throughput data, including microarray and ChIP-seq, have been deposited to GEO and the accession numbers are GSE108144, GSE108145, GSE108146.

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

## Acknowledgements

This work was supported in part by the U.S. the Research Scholar Award RSG-12-085-01 (to J.Y.) from the American Cancer Society, the National Institutes of Health R01CA172384 (to J.Y.), R50CA211271 (to J.C.Z), the Northwestern Prostate SPORE (P50 CA180995), and the Department of Defense Awards #W81XWH-17-1-0578 (to J.Y.) and #W81XWH-17-1-0405 (to J.Y.).

## Author contributions

J.Y. and K.F. conceived and supervised the project. J.Y. and K.F. designed the experiments. K.F., B.Z., and B.S performed the experiments. J.C.Z conducted bioinformatics analysis. J.Y., K.F., and J.C.Z generated the figures and wrote the manuscript. All authors discussed the results, commented on the manuscript, and declared no conflicts of interest.

## Additional information

**Competing interests:** The authors declare no competing interests.

