## [Peer Review File · Nature Communications]

This manuscript has been previously reviewed at another journal that is not operating a transparent peer review scheme. This document only contains reviewer comments and rebuttal letters for versions considered at Nature Communications. Mentions of prior referee reports have been redacted.

POINT-BY-POINT REBUTTAL

Manuscript Number: NCOMMS-18-10732-T

“TRIM28 protects TRIM24 from SPOP-mediated degradation and promotes prostate cancer progression”

Reviewer #1 (Remarks to the Author):

In this manuscript Fong et al. report that TRIM28 prevents wt SPOP-mediated degradation of TRIM24 and enhances TRIM24 and AR signaling. By employing in silico as well as in vitro and in vivo studies they show that TRIM28 expression is upregulated in aggressive human PCa and promotes prostate cancer tumorigenesis. This is a well written and organized study that significantly extends original observations by Groner et al. (Cell 2016) regarding TRIM24 but importantly proposes TRIM28 as a regulator of TRIM24 via interference with SPOP-mediated ubiquitination of TRIM24.

Response: We thank this reviewer for the positive comments regarding our work.

Major comments:

1) The authors suggest that TRIM28 is highly expressed in SPOP WT prostate cancer cell lines. However, it appears that TRIM28 is overexpressed in a wide variety of normal tissues. In SPOPmut tumors do they observe lower expression? This is not directly demonstrated.

Response: This is a great question. To address this question, we re-analyzed TRIM28 expression in SPOP-WT and -Mut tumors in the SU2C dataset (newly added Supplementary Fig. 6c). We observed a trend of increased TRIM28 expression in SPOP-WT tumors, although it is not statistically significant due to large variation of TRIM28 expression in patient samples (text pg17).

2) This study focuses on SPOP WT PCa - via cell line models - where TRIM28 overexpression plays a similar role as SPOPmut with regards to preventing TRIM24 degradation. It is intriguing to consider that overexpression of TRIM28 phenocopies SPOP mutant PCa. As demonstrated in Barbieri et al. (Nat Gen 2012), and more recently in Armenia et al. (Nat Gen 2018) that SPOPmut PCa is associated with either the presence or absence or other distinct genomic alterations. These include genomic loss on chromosome 5 and 6, including loss of CHD1. SPOPmut tumors are often WT for PTEN alterations and do not harbor ETS gene fusions. Therefore, it would be important for the authors to present the TRIM28 overexpressing tumors in this context. What is the role of CHD1?

Response: Recent studies have reported many substrates, in addition to TRIM24, of SPOP. Likewise, TRIM28 has many other functions and is best known as a regulator of DNA damage response. Therefore, as the reviewer stated TRIM28 overexpression plays a similar role as SPOPmut in the context of TRIM24 degradation, but certainly not in all aspects. To address the reviewer's question regarding TRIM28 in relating to the Armenia et al. study, we analyzed TRIM28 expression in the context of various genomic alterations. We did not find any association between TRIM28 level and CHD1 loss, PTEN alteration, or ETS gene fusions (newly added Supplementary Fig. 7e-h), which suggests that the genomic landscape of TRIM28-overexpressing tumor is distinct from that of SPOP-Mut tumors. Interestingly, we found that TRIM28 level is significantly associated with RB1 loss, which is more prevalent in metastatic samples similar as TRIM28 (discussion in pg20-21).

3) The authors address associations with AR signaling and TRIM28 expression. These are important observations. AR signaling was demonstrated in the TCGA to be lowest amongst the ETS fusion PCa cases and highest in the SPOP and FOXA1 tumors (all hormone naïve). The more recent Armenia et al. study puts all the data from a number of genomic studies together revealing that SPOPmut PCa is depleted in CRPC. One possible explanation has to do with increased dependence on AR signaling in

SPOPmut tumors. Could the authors elaborate on if the TRIM28 alterations compensate for this dependency?

Response: We thank the reviewer for this thoughtful comment. We agree that androgen-dependent SPOPmut PCa may be depleted following androgen deprivation therapy, leading to lower incidence in metastatic CRPC. TRIM28 overexpression as observed in CRPC may compensate for this as TRIM28 dramatically enhances AR signaling (Fig.5), which is a key mechanism to prostate cancer resistance to androgen deprivation therapy. This is now discussed in the manuscript (pg20).

4) The correlations with human PCa samples is important. The authors show a good correlation between nuclear staining of TRIM24 and TRIM28. However, they do not specifically show cases with SPOP mutant status. In that instance, they might expect to see variable expression of TRIM28 expression?

Response: We appreciate this valuable point. We reached out to CHTN, who provide us the TMA, but was unsuccessful in obtaining matched DNA samples. Considering that in our TMA there is only 1 case with high TRIM24 but low TRIM28 staining suggesting SPOPmut (Fig. 6e), we thought it might be under powered anyway. Nevertheless, the reviewer's prediction is correct as we indeed observed variable expression of TRIM28 in the SPOPmut tumors in the SU2C dataset (Fig. 6f).

5) Why do the authors select SPOP Y87C mutant expression and not a more common mutation? Or was this the only mutation that showed this association? There are mouse models of SPOP mutations (Blattner et al. (Cancer Cell 2017)). Are the data consistent in these data sets when comparing WT to SPOPmut?

Response: We used Y87C as a representative. But to address the question, we constructed 2 more common SPOP mutants F133V and W131G and show that, like Y87C, they also abolished TRIM24 ubiquitination (newly added Supplementary Fig. 3e). Similarly, SPOP-F133V and W131G expressing prostate cancer cells are also sensitized to TRIM24 inhibitor (newly added Supplementary Fig. 6g-h).

To address the 2nd question, we re-analyzed RNA-seq data from Blattner et al. study (Cancer cell 2017) and found that TRIM28 is in general expressed at a higher level in SPOPwt organoids (newly added Supplementary Fig. 6d). In addition, we found that the TRIM24-AR gene signature tends to be up-regulated in TRIM28-high organoids, supporting our conclusion that TRIM28 is an upstream regulator of TRIM24.

Minor comments:

1) The authors note that "All PCa-associated SPOP mutations discovered thus far are loss-of-function mutations, affecting evolutionarily conserved residues within the MATH domain and disrupting its ability to bind substrates (An et al., 2014; Geng et al., 2014; Theurillat et al., 2014)". Is it also equally plausible that these mutations lead to a change in substrate affinity and therefore are not uniquely "loss or function" but also "change of function"?

Response: Thanks and revised.

2) 19q13.4 has been suggested to be a risk allele in germline studies of PSA screened men (Xu et al. 2012). The authors may want to speculate on the possible associations with AR signaling.

Response: Thanks for this thoughtful comment. We examined the reported risk allele rs103294 within 19q13.4 and found that it is 4.26mb upstream of the TRIM28 promoter. It is unlikely that this allele will affect TRIM28 expression as a mean to increase prostate cancer risk. This is now discussed in the manuscript (pg22).

3) In addition to qPCR analysis of an shRNA screen of the top 10 TRIM24 interactors it would be important to confirm the knockdown on protein level.

Response: Thanks for this suggestion. We now confirmed the knockdown of all 10 TRIM24 interactors at the protein level (newly added Supplementary Fig. 1b).

4) Supplementary Figure 3A shows the different ubiquitination sites of TRIM24. For easier orientation it would be necessary to add the domains and match the colors to Supplementary Figure 3C.

Response: Thanks. We have added the domains and matched the colors in (updated Supplementary Fig 3f-g).

5) Supplementary Figure 7C is missing.

Response: Sorry about that. We added Supplementary Fig. 7c in the revision.

Reviewer #2 (Remarks to the Author):

TRIM proteins have many biological functions including immunity, cell proliferation and differentiation, signal transduction and autophagy. Recent evidence indicates several TRIM proteins serve as regulators in carcinogenesis and transcription. Two of the TRIM proteins, TRIM24 and TRIM28, have been reported to be crucial transcriptional regulators for hormone response such as AR, ER and RAR.

In this paper, the authors showed that TRIM28 regulates the stability and transcriptional activity of TRIM24 via inhibition of SPOP, which is a substrate-recognizing component of Cul3-based complex type E3 ligase. The authors found that TRIM28 physically interacts with TRIM24 and SPOP and that TRIM28 exhibits TRIM24-mediated AR transcriptional activity. Furthermore, the authors showed that TRIM28 knockdown caused the regression of prostate cancers by PCa xenograft models. These findings suggest that TRIM28 is a critical molecule for the AR-mediated pathways and may be a therapeutic target of prostate cancers.

These insights are relevant in understanding the novel roles of TRIM28 in regulation of AR signal in prostate cancers. This manuscript contains some important issues, such as the molecular insight of TRIM28 in the suppression of SPOP in the ubiquitination on TRIM24. Although the experiments have been well performed and the results are solid, there may be overstatements of results that preclude publication of the manuscript in the present form.

Response: We thank this reviewer for the overall positive comments regarding the experiments and results. The main issue raised here is addressed below and we have revised the manuscript carefully to avoid any overstatement of the results.

Specific points

In this paper, it is very important to clarify the mechanism how TRIM28 inhibits SPOP-mediated ubiquitination/degradation of TRIM24. As the authors showed, TRIM28 could not kick out SPOP on TRIM24; therefore, TRIM28, TRIM24 and SPOP may form a trimolecular complex. When SPOP serves as an E3 ligase, SPOP must bind Cul3 and Rbx1. The authors should perform detailed analysis about the molecular bindings among TRIM24, TRIM28, SPOP, Cul3 and Rbx1. TRIM28 may kick out Cul3 and/or Rbx1 from TRIM24-SPOP complex.

Response: We thank the reviewer for this constructive comment. Following the suggestion, we performed pull down of RBX1-TRIM24 or Cul3-TRIM24 complex in cells with or without TRIM28 overexpression. Unfortunately, similar to the case of SPOP, TRIM28 expression can not disrupt TRIM24 interaction with Cul3 and RBX1 (newly added Supplementary Fig. 3b-c).

To further strengthen our conclusion that TRIM28 blocks SPOP-mediated ubiquitination of TRIM24, we examined post-translational modification sites on the TRIM24 protein (www.phosphosite.org) and identified 5 ubiquitination sites: K303, K325, K341, K458, and K1002 (Supplementary Fig. 3f). Out of these, K303, K325, K341, and K458 were previously validated in 293T cells through single-step immunoenrichment of fractionated ubiquitinated peptides and high-resolution mass spectrometry (Wagner et al., 2011). In particular, K341 was reported to be sensitive to MG132 treatment and predicted to be involved in proteasomal degradation. Indeed, we found that TRIM24 mutant with deletion from 300-460aa covering all these predicted ubiquitination sites were no longer ubiquitinated by SPOP (newly added Fig. 3h). We also noted that mutants K341R and K961R, a lysine at TRIM24 c-terminal, remained to be ubiquitinated by SPOP, suggesting that SPOP-mediated ubiquitination of TRIM24 likely occurs at multiple sites within the region, and it is known that nearby lysines can become targeted when the major ubiquitination site is mutated (Treier et al., Cell 1994). Interestingly, d300-460 mutant retained the ability of TRIM24 to bind SPOP, analogous to TRIM28, which inhibits TRIM24 ubiquitination without blocking its interaction with SPOP. These data support that TRIM28 binding to aa100-260 of TRIM24 prevents SPOP-mediated ubiquitination of TRIM24 at nearby lysine residues (within aa300-460) (Supplementary Fig. 3g).

Reviewer #3 (Remarks to the Author):

This paper describes an important novel axis in prostate cancer mediated by TRIM28, via TRIM24 and possibly AR. The study is well conducted and thorough, the methods are sufficiently detailed to allow reproduction and the statistical significance, where appropriate, is indicated. This is an important finding that adds depth to understanding of TRIM protein function. TRIM28 and TRIM24 are related TRIM proteins and the finding that they functionally interact in cancer development will be of broad interest to the wider community. Although it doesn't undermine the message of the paper, the experiments in Figure 3 and Figure S3 require a little clarity, as detailed below.

Response: We thank the reviewer for the positive comments regarding the rigor of our experiments and the importance of our findings. The questions regarding Figure 3 and Fig. 3d are addressed in details below.

Essential revisions:

- Does the MG132-sensitive loss of TRIM24 following TRIM28 depletion, still occur in SPOP-depleted cells? This seems a key experiment to conclude that TRIM28 protects TRIM24 from SPOP-mediated ubiquitination. I'm aware that no SPOP depletions were performed, is this because it is difficult to deplete?

Response: We thank the reviewer for this constructive comment. We now performed TRIM28 and/or SPOP knockdown in DMSO or MG132 treated LNCaP cells. Indeed, SPOP knockdown fully restored TRIM24 protein and TRIM28 knockdown in these cells failed to further increase TRIM24 (newly added Supplementary Fig. 3a). As a control, we also showed that SPOP depletion abolished TRIM24 ubiquitination, which could not be rescued by TRIM28 knockdown (newly added Supplementary Fig. 3d). Together, these data support an SPOP-dependent mechanism of TRIM28 in stabilizing TRIM24 protein.

- Fig. 3D – TRIM24 in the absence of SPOP is required. Otherwise, why does TRIM24 not precipitate in the absence of SPOP (lane 1, empty vector)? It makes it hard to assess this panel and its conclusions if SPOP is required for TRIM24 expression. If SPOP does not decrease the levels of ectopic TRIM24 (as it

does endogenous TRIM24 in panel C), then ectopic TRIM28 has nothing to prevent, and the fact that TRIM28 does not prevent the TRIM24-SPOP interaction might be non-specific.

Response: We apologize for a tiny mistake in the labeling in lane 1 which has overthrown everything. The figure is now correctly labeled (revised Fig. 3d). Briefly, lane 1 contains ectopic Myc-SPOP but not TRIM24-SFB, which explains the absence of flag (TRIM24) and the presence of Myc (SPOP) in WB. TRIM24-SFB was expressed in lane 2, wherein it interacts with Myc-SPOP. This interaction remained in lane 3, wherein HA-TRIM28 was also expressed. All cells were treated with MG132 for 24hrs to prevent TRIM24 degradation and enable the assessment of this ligase-substrate complex.

- If the authors are going to address the ubiquitination of TRIM24 by SPOP, some mass spectrometry to identify sites of TRIM24 ubiquitination might be required. While this is technically challenging, one experiment might be to see whether any ubiquitination sites change upon co-expression, or depletion, of SPOP. SPOP expression does augment TRIM24 ubiquitination (Fig. 3E) so this approach might work. Does lysine mutation decrease ubiquitination of TRIM24 alone, or perhaps in the presence of SPOP? In terms of understanding transcriptional change, a Ub-resistant TRIM24 might be a useful reagent to assist dissecting the mechanisms of PCa.

Response: We thank the reviewer for this thoughtful suggestion. Following this suggestion, we conducted immunoprecipitation of TRIM24-SFB using anti-flag, which targets SFB tag on the c-terminal, in cells with or without SPOP co-expression. The eluted proteins were scanned for potential ubiquitination by mass spectrometry. However, only K961 was found to be ubiquitinated in our mass spectrum (data not shown), probably due to technical challenges associated with ubiquitinated protein fragmentation. In addition, K961 ubiquitination might play cellular roles rather than degradation and indeed its mutation does not block SPOP-mediated ubiquitination of TRIM24 (new Fig. 3h). As detailed in our response to reviewer 2 above, we instead found 300-460aa region of TRIM24 critical for its ubiquitination, which may involve multiple lysine residues.

Minor points

- molecular weight markers should be indicated on gels

Response: Thanks. They are added to the figures now.

- pg 9 states that TRIM28 ‘physically interacts’ with TRIM24. This isn’t shown, as in all experiments a cell lysate is involved. Therefore should acknowledge that TRIM24-TRIM28 binding is not necessarily direct. The ability of TRIM28 to stabilise TRIM24 levels might be enzyme dependent.

Response: Thanks. Agree and revised.

- Fig. 3E – TRIM28 expression does more than overcome SPOP-mediated TRIM24-ubiquitination. What is the assumption here – that TRIM28 also overcomes endogenous SPOP-mediated TRIM24 ubiquitination in the 293Ts? This seems a reasonable assumption given then equivalent loss of TRIM24-ubiquitin by expression of the SPOP mutants in panel F.

Response: Yes, the reviewer is correct that TRIM28 overexpression fully abolished both endogenous and exogenous SPOP-mediated TRIM24 ubiquitination (pg 10).

- Pages 11-12 and Fig S3. Panel B, which bands in the input western blot correspond with the TRIM24 constructs transfected? A negative control lane should be included.

Response: Thanks for the suggestion. We repeated the experiment with an empty vector (negative control) included (newly added Fig. 3g). The bands corresponding to TRIM24 constructs (Flag) are indicated by black arrows.

• Panel 6E – is this a strong correlation? Better to describe it as a positive correlation.

Response: Thanks and revised accordingly (pg 16).

• Fig. S7C missing? **Response:** Sorry about that. We added Supplementary Fig. 7c in the new version

• PMID 28592290: does this need referencing given the authors also uncover an association between TRIM28 and prostate cancer?

Response: Thanks and sorry for missing it. We have now cited PMID 28592290.

Reviewer #4 (Remarks to the Author)

This is a very nice study providing a novel mechanistic understanding of regulation of TRIM24, an oncogenic transcriptional activator, in prostate cancer without SPOP mutation. Using an integrated biochemical and genomic approach, the authors found that TRIM28 is an important upstream regulator of TRIM24. TRIM28 enhances TRIM24 protein stability and facilitates TRIM24 chromatin binding and target gene regulation. Many elements of this manuscript are of exceptional quality and provide very thorough insight into both molecular interactions between TRIM28 and TRIM24 and functional outcomes of such interaction. In addition, it also seems that these results will have a strong potential to impact clinical decision making. Particularly, the authors found that TRIM28 high prostate cancer cells are more sensitive to TRIM24 bromodomain inhibitors. I have just a few points to make that should be addressed before publication is considered.

Response: We thank the reviewer for the positive comments regarding the quality and significance of our study.

1. Figure 5: the authors convincingly demonstrate that TRIM24 enhances AR target gene expression in LNCaP cells. It would be nice to study whether TRIM24 also enhances AR signaling in AR-positive castration-resistant prostate cancer (CRPC) cells.

Response: Thanks for the great suggestion. We repeated the experiment in CRPC cell line C4-2B and found that TRIM24 and TRIM28 are also required for AR target gene expression in CRPC cells (newly added Supplementary Fig. 5c).

2. Figure 6B: Is gene expression of TRIM28 and TRIM24 correlated in metastatic CRPC patients?

Response: We actually observed a positive correlation ($r=0.669$) between TRIM28 and TRIM24 mRNA levels in metastatic CRPC samples (see figure on the right). As our study focuses on TRIM28 regulation of TRIM24 at protein level, this data appears irrelevant and we did not include it in the manuscript so as to avoid confusion.

3. Figure 6H: Are TRIM28-high CRPC cells also more sensitive to TRIM24-C34 inhibitor?

Response: The reviewer is correct. We repeated the experiment in CRPC line C4-2B and found that TRIM28-high CRPC cells are also more sensitive to TRIM24-C34 inhibitor as compared to TRIM28-low CRPC cells (newly added Supplementary Fig. 6g-h).

4. Figure 7D. Does TRIM28 silencing affect mice body weight?

Response: TRIM28 silencing have no significant effect on mice body weight (newly added Supplementary Fig. 7d).

Reviewers' Comments:

Reviewer #1:

Remarks to the Author:

The authors have adequately addressed my comments.

Reviewer #2:

Remarks to the Author:

Now it is acceptable for publication in Nature communications.

No more comments.

Reviewer #3:

Remarks to the Author:

The authors have satisfactorily addressed the concerns I raised and I would be happy to see this manuscript carried forward into a publication.

Reviewer #4:

None

POINT-BY-POINT REBUTTAL

Manuscript Number: NCOMMS-18-10732-A

“TRIM28 protects TRIM24 from SPOP-mediated degradation and promotes prostate cancer progression”

Reviewer #1 (Remarks to the Author):

The authors have adequately addressed my comments.

Response: We thanks the reviewer for supporting our manuscript for publication.

Reviewer #2 (Remarks to the Author):

Now it is acceptable for publication in Nature communications.

No more comments.

Response: We thanks the reviewer for supporting our manuscript for publication.

Reviewer #3 (Remarks to the Author):

The authors have satisfactorily addressed the concerns I raised and I would be happy to see this manuscript carried forward into a publication.

Response: We thanks the reviewer for supporting our manuscript for publication.

Reviewer #4

Commented to the editors only that Reviewer #4 is supportive of publication.

Response: We thanks the reviewer for supporting our manuscript for publication.